

# Formation of highly oxygenated organic molecules from aromatic compounds.

Ugo Molteni[1], Federico Bianchi[1-2], Felix Klein[1], Imad El Haddad[1], Carla Frege[1], Michel J. Rossi[1], Josef Dommen[1], Urs Baltensperger[1,*]

[1]Laboratory of Atmospheric Chemistry, Paul Scherrer Institute, CH-5232 Villigen, Switzerland
[2]Department of Physics, University of Helsinki, 00014 Helsinki, Finland

*Correspondence to*: Urs Baltensperger (urs.baltensperger@psi.ch)

## Abstract

Anthropogenic volatile organic compounds (AVOC) often dominate the urban atmosphere and consist to a large degree of aromatic hydrocarbons (ArHC), such as benzene, toluene, xylenes, and trimethylbenzenes, e.g. from handling and combustion of fuels. These compounds are important precursors for the formation of secondary organic aerosol. Despite their recognized importance as atmospheric reactants, the formation of highly oxygenated molecules (HOMs) in the gas phase leading to (extremely) low volatility compounds has not been studied in the past. Here we show that oxidation of aromatics with OH leads to a subsequent autoxidation chain reaction forming HOMs with an O:C ratio of up to 1.09. This is exemplified for five single-ring ArHC (benzene, toluene, o-/m-/p-xylene, mesitylene (1,3,5-trimethylbenzene) and ethylbenzene), as well as two conjugated polycyclic ArHC (naphthalene and biphenyl). We present the identified compounds, differences in the observed oxidation patterns and discuss mechanistic pathways. We hypothesize that AVOC may contribute substantially to new particle formation events that have been detected in urban areas.

## 1 Introduction

Volatile organic compounds (VOCs) from biogenic and anthropogenic sources are the precursors of atmospheric oxidation products in the gas and particle phase. While global biogenic VOC (BVOC) emissions are a factor of 10 higher than the emissions of anthropogenic VOCs (AVOCs), the latter often dominate in the urban atmosphere (Atkinson and Arey, 2003). It was shown recently that atmospheric oxidation products of BVOCs, such as the monoterpene alpha-pinene, include highly oxygenated molecules (HOMs) through an autoxidation mechanism (Crounse et al., 2012, 2013; Ehn et al., 2014). The first step is a reaction of either OH free radicals or ozone with the VOC. After addition of $O_2$ to the carbon-centered radical site the $RO_2\cdot$ radical can isomerize by intra-molecular hydrogen abstraction to form a new carbon-centered radical (QOOH) (Crounse et al., 2012, 2013; Ehn et al., 2014). Further $O_2$-addition/isomerization sequences result in HOMs bearing several hydroperoxy groups. This autoxidation mechanism is supported by various experimental studies which used biogenic precursors, i.e. monoterpenes, sesquiterpenes, isoprene, and structural surrogates of these and computer simulations (Berndt





et al., 2015; Jokinen et al., 2014, 2015; Kurtén et al., 2015; Mentel et al., 2015; Praplan et al., 2015; Richters et al., 2016; Rissanen et al., 2014, 2015). HOMs of those compounds were found to initiate new particle formation and substantially contribute to early particle growth, which is important for the survival of newly formed particles and their ability to form cloud condensation nuclei, CCN (Bianchi et al., 2016; Kirkby et al., 2016; Tröstl et al., 2016). CCN can impact climate via their influence on cloud properties nowadays and in the pre-industrial period (Carslaw et al., 2013; Gordon et al., 2016).

AVOCs are comprised of a high fraction of aromatic hydrocarbons (ArHC), such as benzene, toluene, xylenes, and trimethylbenzenes, which are released from handling and combustion of fuels (Atkinson and Arey, 2003),  and are important precursors for the formation of secondary organic aerosol (SOA) (Bruns et al., 2016; Li et al., 2016; Metzger et al., 2010). The OH radical is the preponderant atmospheric oxidant for ArHC except for phenols or substituted ArHC with non-aromatic double bonds where ozone and the $NO_3$ radical play a relevant role (Calvert et al., 2002). The addition of the OH radical to the aromatic ring results in the formation of a hydroxycyclohexadienyl-type radical (Bohn, 2001; Molina et al., 1999). Despite the fact that ArHC·OH adducts under atmospheric conditions react with $O_2$ to yield peroxy radicals (Calvert et al., 2002; Glowacki and Pilling, 2010; Suh et al., 2003) and the recognized importance of ArHC in the photochemical production of ozone and SOA, it is not known if ArHC oxidation also yields HOMs (Birdsall and Elrod, 2011; Li and Wang, 2014; Pan and Wang, 2014; Wang et al., 2013; Zhang et al., 2012). This could be linked to the fact that the relevant processes were neither accessible by flash photolysis nor by smog chamber experiments (Glowacki and Pilling, 2010) and instrument limitations. In many studies no carbon balance could be reached and generally only about 50% of the carbon reacted was identified as products (Calvert et al., 2002). When aromaticity is destroyed by OH-addition non-aromatic double bonds are formed, thus representing highly reactive products, which is a peculiar behaviour not observed in other classes of VOC (Calvert et al., 2002). This behaviour makes the investigation of ArHC oxidation more complex.

Here we present product distributions in terms of molecular masses for ArHC HOMs upon reaction of aromatic compounds with OH radicals, based on measurements with a nitrate chemical ionization atmospheric pressure interface time of flight mass spectrometer (CI-APi-TOF) (Ehn et al., 2014; Jokinen et al., 2012; Kürten et al., 2011). Potential pathways and a possible mechanism for the formation of HOMs from aromatic compounds are discussed.

## 2 Experimental section

### 2.1 Flow tube

Five single-ring ArHCs: benzene (Merck), toluene (VWR Chemicals), a mixture of o-/m-/p-xylene isomers (Merck), mesitylene (1,3,5-trimethylbenzene) (Fluka), ethylbenzene (Fluka), as well as two polycyclic ArHCs naphthalene (Fluka) and biphenyl (Sigma-Aldrich) were investigated in a flow tube (Table 1). The experimental set-up is shown in Figure 1. Zero air from a pure air generator (Aadco Instruments, Inc., Cleves OH, USA) was used. A 104-cm long Pyrex glass tube of 7.4 cm diameter described previously (Pratte and Rossi, 2006) was used as a flow tube. Vapours of the aromatic compounds were generated from a glass vial, and collected by a stream of zero air (1.1 L min$^{-1}$) via a glass capillary for liquid



compounds (and from a flask flushed with the same stream of zero air for solid compounds). To generate OH free radicals, zero air (7 L min$^{-1}$) was passed through a Nafion humidifier (Perma Pure) fed with ultra-pure water, and was then irradiated

by an excimer lamp at 172 nm (7.2 eV) (Kogelschatz, 1990, 2012; Salvermoser et al., 2008). The Xe excimer lamp has a coaxial geometry and consists of a tubular quartz cell which surrounds a quartz flow tube (outer diameter 10 mm) which is used as OH radical generator. Previous works (Bartels-Rausch et al., 2011) used this set up for HO$_2$ radical generation. Subsequently, the air stream with the OH free radicals was combined at an angle of 90 degrees with the reagent flow containing the aromatic vapours before entering the flow tube, initiating the oxidation reaction. This experimental set-up

avoids any potential bias due to exposure of ArHC vapours to UV radiation (Jain et al., 2012; Peng et al., 2016). This mixture (total 8.1 L min$^{-1}$) was injected into a laminar sheath flow of 6.7 L min$^{-1}$ zero air at the inlet of the flow tube. The residence time in the flow tube was 20 sec. All experiments were performed at 25° C.

## 2.2 Instruments

The concentration of the ArHC precursors and D9-butanol as an OH tracer was measured at the exit of the flow tube with a

proton-transfer-reaction time of flight mass spectrometer (PTR-TOF-MS) (Jordan et al., 2009) when the excimer lamp to generate OH free radicals was switched off and on. A nitrate chemical ionization atmospheric pressure interface time of flight mass spectrometer (CI-APi-TOF) (Ehn et al., 2014; Jokinen et al., 2012; Kürten et al., 2011) measured the chemical composition of the HOMs that were formed via OH free radical oxidation of the aromatics. HOMs were detected either through acid-base reaction or adduct formation with a nitrate ion according to the scheme:

$$HOM + NO_3^- \cdot (HNO_3)_n \rightarrow HOM \cdot NO_3^- + (HNO_3)_n \quad n=0\text{-}2 \quad (R1)$$
$$HOM + NO_3^- \cdot (HNO_3)_n \rightarrow HOM^- + (HNO_3)_{n+1} \quad n=0\text{-}2 \quad (R2)$$


Trifluoroacetic acid (monomer and dimer) was detected as major contaminant in the CI-APi-TOF spectra. We identified the Nafion humidifier membrane as the source of fluorinated organic compounds.

The OH free radical concentration was estimated from two separate experiments using D9-butanol following the method of Barmet et al. (2012). From the D9-butanol signal with excimer lamp on and off we obtained an average OH concentration of

$(1.9\pm0.4)\ 10^8$ molecules cm$^{-3}$. Ozone, produced in the excimer irradiated region as a side product of OH generation, was measured to be about 140 ppbv at the exit of the flow tube and is therefore not expected to play a significant role in the oxidation of ArHC in flow tube experiments.

## 3 Results and discussion

### 3.1 Comparison of HOMs from different ArHC

The oxidation products of the OH reaction with each of the five single-ring and two polycyclic ArHCs were measured at the exit of the tube using the CI-APi-TOF. Table 1 lists the initial concentration and the reaction rate constant of them with the



OH radical and ozone, respectively. All investigated compounds yielded HOMs in a range between 0.3 and 4 % of the reacted ArHC. They were detected either as adducts with a nitrate ion ($NO_3^-$) or as deprotonated ions. Appendix A presents HOMs peak lists for all the ArHC compounds: for each compound we report the first $n$ peaks that sum up to 80% of the total

detected signal of HOMs. Figure 2 displays the mass spectra obtained from the monocyclic aromatics. In the mass-to-charge ($m/z$) range 130 – 365 thomson (Th; 1 Th =1 Da $e^{-1}$, where $e$ is the elementary charge), the oxidation products contain the carbon skeleton of the precursor (monomer region), while in the $m/z$ range 285 – 540 Th the number of carbon atoms is doubled (dimer region). The lower end of the peak sequence (which for the benzene experiment corresponds to the oxidation product with formula $C_6H_6O_5(NO_3)^-$) is shifted by differences of 14 Th ($CH_2$) each from benzene via toluene and

xylene/ethylbenzene to mesitylene due to the additional methyl/ethyl groups. In general, a series of peaks with a mass difference of two oxygen atoms can be seen in the monomer as well as the dimer region. At each oxygen addition a few peaks are observed because oxidation compounds with the same carbon and oxygen number but different hydrogen number were observed. These peaks can be attributed to closed shell or radical compounds based on the number of hydrogen (even or odd).

HOMs from naphthalene and biphenyl are presented in Figure 3. Monomers, dimers, trimers, and tetramers are observed, and even pentamers for biphenyl. While some of the dimers may have been formed by $RO_2\cdot - RO_2\cdot$ reactions, most of the higher n-mers are probably bonded by Van der Waals interactions, similar to the mechanism for biogenic HOMs (Kirkby et al., 2016; Tröstl et al., 2016). Clusters with $m/z \geq 800$ Th might already be detected as particles with a mobility diameter $d \geq$ 1.5 nm (Kulmala et al., 2013).

In Table 2 we summarize the general features of the peak distribution of monomers, dimers and $n$-mers, as well their O:C ratios. The values given in the table cannot be considered to be absolute values, since we do not know the transmission function of the mass spectrometer. Thus, the dimer/monomer ratio might be different. However, the given values may be a good proxy of the relative behaviour of the product distribution of the different aromatic compounds. Most of the identified peaks (77-94%) were detected as adduct with $NO_3^-$. The integrated signal intensity in the monomer region makes up 61 to

80% of the total detected ArHC products signal for the monocyclic ArHCs and 34-52% for the double-ring compounds. A further analysis of HOMs from monocyclic ArHC shows an increase in the dimer fraction which coincides with an increase in the methyl/ethyl substituents as follows: benzene (20%), toluene (29%), ethylbenzene (31%), xylene (35%), mesitylene (39%). If we assume that the $HO_2$ concentration is similar, then the branching ratio of $RO_2\cdot + RO_2\cdot$ to dimer compared to the other reaction channels is higher for the more substituted aromatics. Monomers as well as dimers are highly oxygenated,

even though the molecular oxygen-to-carbon (O:C) ratio is 20-30% higher for the monomers compared to the dimers. Single-ring ArHC monomers have on average an O:C ratio of 0.94 (0.50 for the double-ring ArHC) while monocyclic ArHC dimers have on average an O:C ratio of 0.67 (0.32 for the double-ring ArHC). This may be due to the dimer formation mechanism itself, which is thought to be the formation of a peroxide C-O-O-C bond which involves elimination of molecular oxygen (Kirkby et al., 2016; Wallington et al., 1992). Additionally, more oxygenated radicals have a higher probability to

undergo an auto-termination radical reaction compared to a radical-radical recombination ($RO_2\cdot + RO_2\cdot$ or $RO_2\cdot + HO_2\cdot$).





Furthermore, less oxygenated products are not quantitatively detected by the CI-APi-TOF (Berndt et al., 2015; Hyttinen et al., 2015) but such radicals are nevertheless taking part in the dimer formation. Substantially lower O:C ratios are found for naphthalene and biphenyl, whereby the trend between the monomers and the dimers and higher order clusters is the same as for the single-ring ArHCs. The lower O:C ratio is probably due to the fact that the second aromatic ring remains and does not

allow for extensive autoxidation.

Figure 4 shows the contribution of the most abundant identified HOMs to 80% of the total signal. The chemical composition of the observed monomers, dimers and radicals for each precursor is presented in Appendix B (Figure B-1 to Figure B-7). It is seen that in the series benzene, toluene, xylene, mesitylene the number of HOMs needed to sum up 80% of the total signal decreases (except for ethylbenzene). The increasing number of methyl groups appears to influence the oxidation pathway

and leads to less HOM products. Ethylbenzene shows the highest number of HOMs. These also include monomers with 7 carbon atoms as well as dimers with an unexpectedly low number of hydrogen atoms (20 instead of 22) (Figure B – 3). This could indicate the occurrence of different pathways due to the ethyl group, a chemistry less bounded to the aromatic ring which implies an initial hydrogen abstraction step by the OH radical. Together with ethylbenzene also benzene and naphthalene, the two not substituted ArHC tested, present dimers with an unexpected low hydrogen number (12 instead of

14 and 16 instead of 18). Biphenyl also shows an unexpectedly low number of hydrogen atoms for some of the HOM monomers detected. This feature, highlighted in the four above-mentioned compounds, it turns out to be a minority in terms of peaks detected and relative peak intensity.

## 3.2 ArHC HOMs formation mechanism

A generalized mechanism that may explain the formation of these highly oxygenated compounds from ArHCs by OH addition is exemplified for mesitylene in Figure 5. This mechanism is also applicable to the other ArHCs tested. An OH free radical attack on alkyl-substituted arenes is thought to either abstract a hydrogen atom from an $sp^3$ hybridized carbon, or to add to the aromatic ring. Starting from a generic aromatic compound with formula $C_xH_y$, hydrogen abstraction results in a $C_xH_{y-1}$ radical while OH addition results in a radical with the formula $C_xH_{y+1}O_1$. If we allow both initial intermediate products

to proceed via autoxidation by formal addition of $O_2$, we expect radicals with the composition $C_xH_{y-1}O_z$ (initial hydrogen abstraction) and $C_xH_{y+1}O_z$ (initial OH addition) to be formed, where z denotes the number of oxygen atoms, even number (ze) in the former case and odd number (zo) in the latter case. This addition of molecular $O_2$ increases the mass of the compounds by 32 Da resulting in the propagation of a radical with an odd number of oxygen atoms. This can be seen by *m/z* shifts of 32 Th in the mass spectra. For the ArHCs tested we do not observe radicals with the formula $C_xH_{y-1}O_{ze}$, owing to the

fact that hydrogen abstraction is a minor pathway (with e.g. a branching ratio of 7% for toluene according to the Master Chemical Mechanism MCM 3.3.1 (Jenkin et al., 2003), which yields products like benzaldehyde and benzyl alcohol). For mesitylene (Figure 5), the OH-adduct and the first RO₂· radical (HO-$C_xH_y$OO·) cannot be detected with the nitrate CI-APi-TOF (Hyttinen et al., 2015) and are reported in grey in Figure 5. More highly oxygenated RO₂· radicals with the formula




$C_9H_{13}O_{5-11}$ were however found, with the highest intensity for $C_9H_{13}O_7$ (3% of the sum of the identified HOMs). In addition

to radicals with an odd oxygen number, radicals with an even oxygen number of molecular formula $C_xH_{y+1}O_{ze}$ were observed (Figure 5). These radicals are likely produced via $RO_2 \cdot + RO_2 \cdot$ (or $RO_2 \cdot + HO_2 \cdot$), involving the formation of an alkoxy radical intermediate (Kirkby et al., 2016; Lightfoot et al., 1992; Orlando and Tyndall, 2012; Vereecken and Peeters, 2009) according to:

$$
\begin{array}{llll}
ROO \cdot + R'OO \cdot & \rightarrow & RO \cdot + R'O \cdot + O_2 & (R3a) \\
& \rightarrow & ROH + R'_{-H}O + O_2 & (R3b) \\
& \rightarrow & ROOR' + O_2 & (R3c) \\
ROO \cdot + HOO \cdot & \rightarrow & ROOH + O_2 & (R4a) \\
& \rightarrow & RO \cdot + \cdot OH + O_2 & (R4b)
\end{array}
$$


These alkoxy radicals (R3a) may isomerize to an alcohol by internal H-abstraction forming a carbon centred radical, which can again uptake an oxygen atom and follow the autoxidation route. The peroxy radicals of this reaction channel have the formula $C_xH_{y+1}O_{ze}$ (see Fig.5). Besides the formation of alkoxy radicals recombination can also lead to a carbonyl and alcohol species (R3b) with the formulae $C_xH_yO_z$ and $C_xH_{y+2}O_z$. The discrepancy between the intensity of the peaks with

formula $C_xH_yO_z$ and the peaks with formula $C_xH_{y+2}O_z$ can be ascribed to the presence of compounds resulting from the recombination of $RO_2 \cdot$ with $HO_2 \cdot$ in the latter class(R4a). The formation of ROOR (R3c) corresponds to $C_{2x}H_{2y+2}O_z$ dimer formation with z being even or odd, depending on the combination of the reacting peroxy radicals. We also detected free radicals and closed-shell molecules with an unexpectedly high number of hydrogen atoms, with the formulae $C_xH_{y+(3,5)}O_z$ and $C_xH_{y+(4,6)}O_z$, respectively. For mesitylene (Fig. 5), radicals with the formula $C_9H_{15}O_{7-11}$ were identified, with the highest

signals found for $C_9H_{15}O_7$ (1%), and $C_9H_{15}O_8$ (2%). These compounds are likely formed by a second OH addition as discussed further below. Monomer closed-shell molecules were detected as $C_9H_{12}O_{5-11}$ (4%), $C_9H_{14}O_{4-11}$ (25%), and $C_9H_{16}O_{5-10}$ (12%). We assume that the $C_9H_{12}O_{5-11}$ molecules derive from the first radical generation ($C_9H_{13}O_{5-11}$) and the $C_9H_{16}O_{5-10}$ molecules from a second OH attack ($C_9H_{15}O_{7-11}$). The $C_9H_{14}O_{4-11}$ molecules may be produced from either the first or the second OH attack. However further investigation is required to test these hypotheses. We also want to point out that the

relative signal intensities may be biased by the nitrate clustering properties and do not necessarily reflect the actual distribution of compounds. Similarly, the compounds with a lower than expected H-atom number could have been formed by a H-abstraction from first generation products with formula $C_xH_yO_z$.

The recombination of two peroxy radicals may lead to a covalently-bound peroxy-bridged dimer. We observed three classes of such products (Figure 5): i) from the recombination of two first-generation radicals (13 hydrogen atoms each) with the

molecular formula $C_{18}H_{26}O_{8-13}$ (30% of the total intensity), ii) from the recombination of a first generation radical with a second generation radical (13 + 15 hydrogen atoms) with formula $C_{18}H_{28}O_{9-12}$ (3% of the total signal), and iii) from the recombination of two radicals from the second generation (15 + 15 hydrogen atoms), where only one compound was identified ($C_{18}H_{30}O_{11}$, 1%).



Some identified monomer and dimer peaks belong to oxygenated molecules with less carbon atoms than the respective

precursor. This is likely the result of a fragmentation process. HOMs with less C atoms than the parent molecule have also been previously described from terpene precursors via CO elimination (Rissanen et al., 2014, 2015). The aromatics show mostly much less H-atoms than the terpenes after fragmentation. This indicates that a methyl group is lost, e.g. as formaldehyde. As mentioned above, we hypothesize that the $C_xH_{y+(3,5)}O_z$ radicals and $C_xH_{y+(4,6)}O_z$ molecules may have formed by multiple OH attacks in our reactor. This is possible when the second OH attacks a product molecule that contains two

hydrogen atoms more than the parent molecule. To allow for the addition of a second OH free radical these first generation closed-shell molecules must still contain a double bond in their structure. A third OH attack is observed only for some compounds; the mechanism will likely proceed in a similar way.

An explicit mechanism after OH addition for a possible pathway of the aromatic autoxidation is suggested in Figure 6 for up to seven oxygen atoms. Compared to aliphatic compounds, for which the autoxidation mechanism proceeds by hydrogen

abstraction and formation of a hydroperoxyalkyl radical that reacts with molecular oxygen, aromatic compounds - once their aromaticity is lost - can form a conjugated radical in the allylic position for a subsequent molecular oxygen attack and peroxy radical formation (Baltaretu et al., 2009; Birdsall and Elrod, 2011; Pan and Wang, 2014). The peroxy radical then can either abstract a hydrogen atom, when possible, or attack the double bond producing a second stabilized allylic radical forming an endocyclic $O_2$ bridge. This process can continue up to seven oxygen atoms ($C_9H_{13}O_7$), which is the species

detected at relatively high intensity (3.5%). This mechanism varies among the ArHC tested. Aromatics with a lower number of methyl/ethyl substituents seem to form radicals with a higher number of oxygen (i.e., up to 9-11 atoms, Appendix B). However, it appears that a single ring ArHC can host up to a maximum of 11 oxygen atoms and a ring opening step seems to be a requirement to reach such a high O:C ratio. In Figure 6 we hypothesize possible branching channels where this may happen. A peroxy radical recombination can form an alkoxy radical which can decompose by a C-C bond cleavage yielding a

carbonyl group and a carbon centred radical. Such a ring opening step was already proposed for α-pinene to explain the high O:C ratio (Kurtén et al., 2016). Termination reactions of those alkoxy radicals can also form molecules containing still double bonds which can further react with OH radicals leading to compounds with four hydrogen atoms more than the precursor.

As mentioned above naphthalene and biphenyl, despite the polycyclic skeleton, do not show a radically different behaviour

compared to the single-ring ArHCs. The maximum number of oxygen atoms that their monomer HOMs can host is 10 for naphthalene and 11 for biphenyl. Biphenyl seems to compare with its single ring analogue benzene. $C_6H_8O_5$ and $C_{12}H_{12}O_5$ are the strongest peaks indicating that the oxidation of one benzene ring in biphenyl proceeds in a similar way. Similarly, the strongest dimer is $C_{12}H_{14}O_8$ for benzene and $C_{12}H_{22}O_8$ for biphenyl, respectively. Compounds with extra-high H-atoms are more frequently for biphenyl, which is expected as there is a second reactive aromatic ring remaining after (auto)-oxidation

of the first one. Thus, a second OH attack is easily possible. Naphthalene seems to take up less oxygen than the other compounds, showing the maximum signal intensity at 4-5 oxygen atoms for monomers and only 4-6 for dimers. This may





indicate that not both rings can easily be autoxidized in one step. It is also interesting to note that compounds from a second OH attack do not show a strong increase of the oxygen content, neither for the single nor for the double ring ArHCs.

## 4 Conclusions

All tested compounds yielded HOMs and we conclude that this is a common feature of aromatic compounds. Similar to the oxidation process that yields HOMs from terpenes the oxidation process of ArHC yields highly oxygenated compounds containing the carbon skeleton of the precursor (monomers) as well as twice as many carbons (dimers). It is known from previous studies that ArHC are able to add molecular oxygen to the molecule after OH addition forming an oxygen-bridged bicyclic radical. Our measurements of highly oxygenated compounds up to eleven oxygen atoms in a monomer reveal that

an autoxidation radical chain reaction occurs by adding several more oxygens to the initially formed radical. This may happen by further addition of oxygen to the allylic resonance-stabilized radical and formation of oxygen bridges up to a peroxy radical of 7 oxygen. The autoxidation chain may also proceed after a ring opening intermediate step. Even though the autoxidation of ArHCs will lead to different chemical compounds compared to HOMs form terpenes we expect similar chemical and physical characteristics such as functional groups or volatility. In both cases extremely low volatility highly

oxygenated dimer species are formed, which may play an important role in new particle formation.

Recent studies (Nakao et al., 2011; Schwantes et al., 2016) suggest a mechanism where the initial step is the formation of the phenolic equivalent ArHC followed by additional oxidation steps yielding "polyphenolic" structures with high O:C ratio. Literature data are showing varying yields for the conversion of arenes to phenols via the OH radical addition and H elimination. According to MCM 3.3.1 (Jenkin et al., 2003) benzene and toluene have quite high phenol yields

(approximately 50 and 20 %, respectively) while mesitylene shows a rather small yield (4%). In our experiments we do not detect such a difference in the HOMs formation linked to phenol formation yields. We therefore believe that the formation of these ArHC HOMs is a separate oxidation process. Some of the ArHC HOMs identified here from the oxidation of ArHC correspond to the HOMs formulae identified by Bianchi et al. (2016) during winter time nucleation episodes at the Jungfraujoch High Altitude Research Station.

Our findings can help in explaining the missing carbon balance in ArHC oxidation experiments. Furthermore, the fact that the oxidation of aromatic compounds can rapidly form HOMs of very low volatility makes these potential contributors in nucleation and particle growth episodes observed in urban areas where these ArHC are abundant which makes ArHC a potential contributor in nucleation and particle growth episodes observed in urban areas where AVOCs are thought to play a key role (Stanier et al., 2004; Wang et al., 2015; Xiao et al., 2015; Yu et al., 2016).






**Acknowledgments.**

We thank Prof. Dr. Markus Ammann for the laboratory equipment, Kilic Dogushan for technical support, Simone Maria Pieber and Dr. Christopher Robert Hoyle for scientific discussions and comments on the manuscript. The tofTools team is
acknowledged for providing tools for mass spectrometry analysis. This work was supported by the Swiss National Science Foundation (20020_152907 / 1).



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



**Tables**

**Table 1**

Initial concentrations of precursors and reaction rate coefficients. The mixing ratio was determined at the exit of the flow
tube when the excimer lamp (OH generation) was switched off.

| Compound | Concentration (molecules cm$^{-3}$) | $k_{OH}$ ($10^{-12}$ cm$^3$ molecules$^{-1}$ s$^{-1}$) | $k_{O3}$ ($10^{-17}$ cm$^3$ molecules$^{-1}$ s$^{-1}$) |
|---|---|---|---|
| Benzene (C$_6$H$_6$) | 9.85 $10^{13}$ | 1.22 | < 1 $10^{-3}$ |
| Toluene (C$_7$H$_8$) | 1.97 $10^{13}$ | 5.63 | < 1 $10^{-3}$ |
| Ethylbenzene (C$_8$H$_{10}$) | 1.13 $10^{13}$ | 7.0 | < 1 $10^{-3}$ |
| (o/m/p)-xylene (C$_8$H$_{10}$) | 2.95 $10^{12}$ | 13.6/23.1/14.3 | < 1 $10^{-3}$ |
| Mesitylene (C$_9$H$_{12}$) | 2.46 $10^{12}$ | 56.7 | < 1 $10^{-3}$ |
| Naphthalene (C$_{10}$H$_8$) | 2.95 $10^{13}$ | 23.0 | < 0.02 |
| Biphenyl (C$_{12}$H$_{10}$) | 4.43 $10^{13}$ | 7.1 | < 0.02 |

Reference for the *k*-rates: (Atkinson and Arey, 2003)





**Table 2**

Summary of HOM characteristics. For each of the 7 compounds the fractional distribution of the signal is presented. For monocyclic compounds the distribution comprises monomers and dimers, for naphthalene and biphenyl monomers, dimers, trimers and tetramers are reported. These values are not quantitative as the instrument cannot be calibrated for such compounds. For each band the weighted arithmetic means of the O:C ratio are reported in parentheses. The fraction of the identified peaks as adduct with $NO_3^-$ is given in the last column.

| Compound | Bands distribution | | | | Adduct (HOM·$NO_3^-$) |
|---|---|---|---|---|---|
| | Monomer (O:C) | | Dimer (O:C) | | |
| Benzene ($C_6H_6$) | 0.80 (1.08) | | 0.20 (0.91) | | 0.91 |
| Toluene ($C_7H_8$) | 0.71 (1.09) | | 0.29 (0.75) | | 0.94 |
| Ethylbenzene ($C_8H_{10}$) | 0.69 (0.86) | | 0.31 (0.62) | | 0.83 |
| (o/m/p)-xylene ($C_8H_{10}$) | 0.65 (0.78) | | 0.35 (0.57) | | 0.92 |
| Mesitylene ($C_9H_{12}$) | 0.61 (0.81) | | 0.39 (0.49) | | 0.92 |
| | Monomer (O:C) | Dimer (O:C) | Trimer (O:C) | Tetramer (O:C) | |
| Naphthalene ($C_{10}H_8$) | 0.34 (0.55) | 0.64 (0.29) | 0.02 (0.34) | 0.01 (0.28) | 0.84 |
| Biphenyl ($C_{12}H_{10}$) | 0.52 (0.44) | 0.43 (0.35) | 0.04 (0.29) | 0.01 (0.32) | 0.77 |






**Figures**

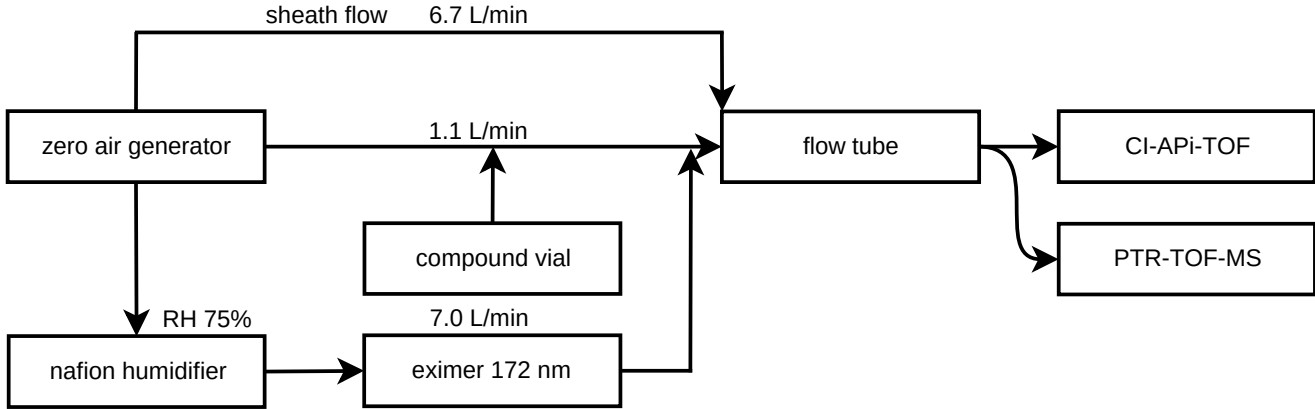

Figure 1. Experimental set-up. Zero air from a pure air generator is split into 3 flows. A sheath flow of 6.7 L min$^{-1}$. An air stream of 1.1 L min$^{-1}$ collects vapours from a reagent compound vial and is then mixed with a humidified air stream of 7 L min$^{-1}$ (RH 75%) which carries OH free radicals generated through irradiation at 172 nm.





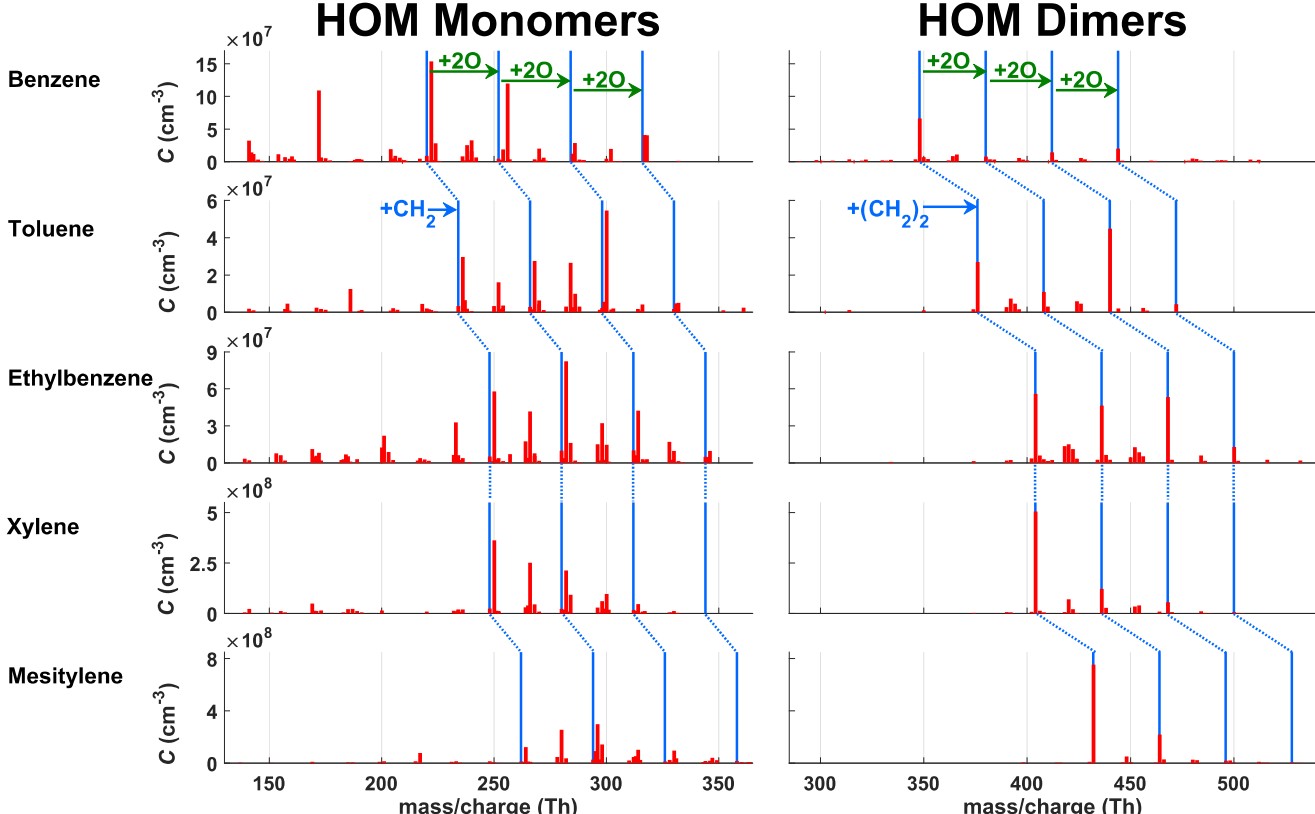

Figure 2. HOMs from 5 monocyclic ArHC (benzene, toluene, ethylbenzene, xylenes, mesitylene). HOM monomers have the same number of carbon atoms as the precursor while HOM dimers have twice as many. Green arrows in the benzene panel show a sequence of peaks separated by a mass corresponding to 2 oxygen atoms. This may be connected to the autoxidation mechanism which proceeds through addition of $O_2$ molecules. The same sequence is seen in the other ArHCs as indicated by the blue lines. The initial peak and the corresponding sequence of the 5 single-ring ArHC are shifted by a $CH_2$ unit due to the different substituents (blue arrow in the toluene panel). HOM peaks (in red) are not always aligned to the blue lines doe to the unequal prevalence of HOMs with different hydrogen atoms (n, n+2, n+4). Note the different peak intensity patterns observed for the different chemical compounds, even for xylene and ethylbenzene, i.e., molecules with the same chemical formula.





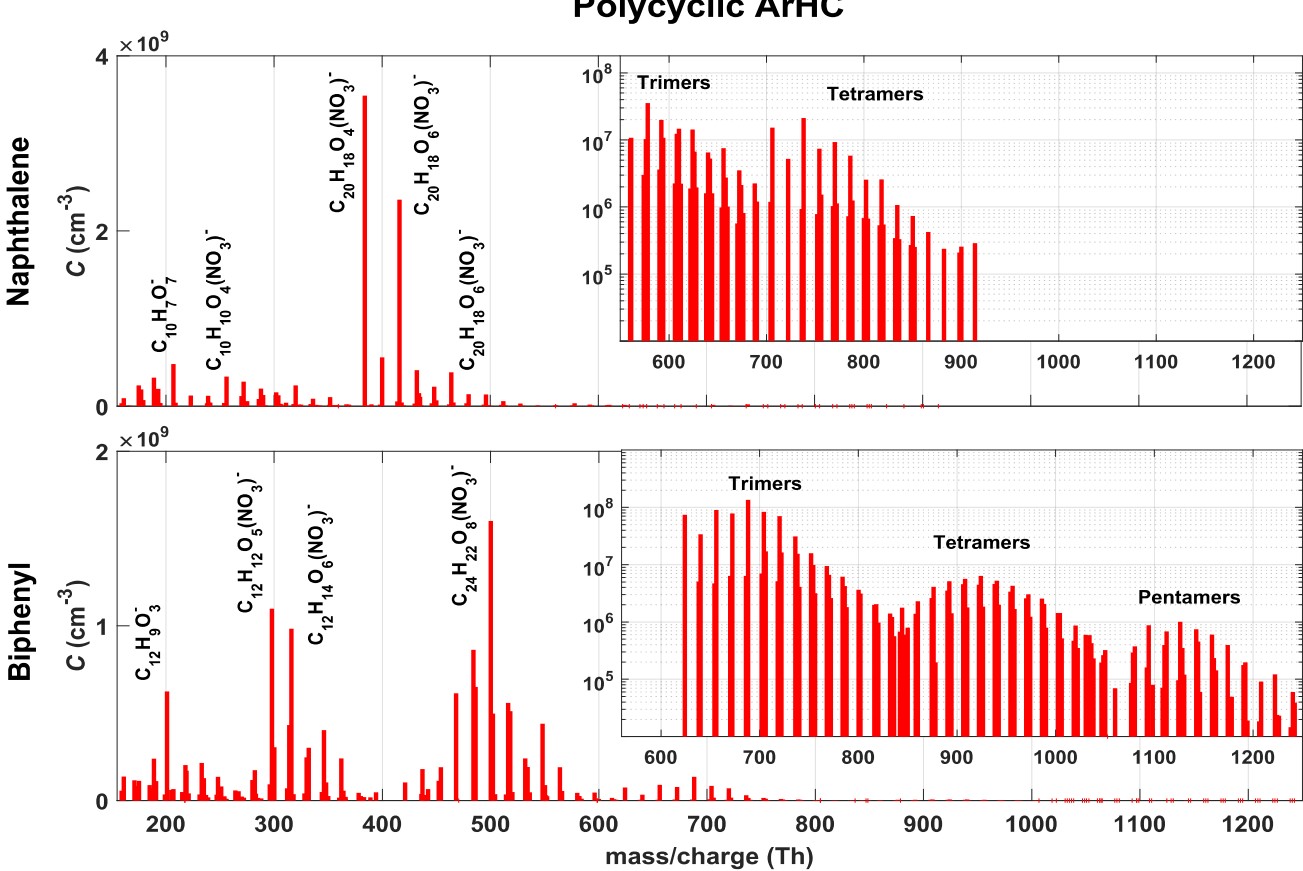

Figure 3. Mass spectra of HOMs from the bicyclic ArHC naphthalene and biphenyl. The chemical composition of some representative peaks is displayed. Due to the high concentrations the nitrate CI-APi-TOF was able to also detect the HOMs clusters up to the pentamer and also retrieve their chemical formula (see inserts).


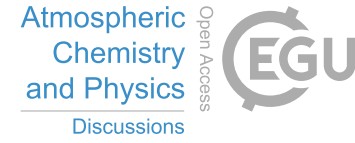



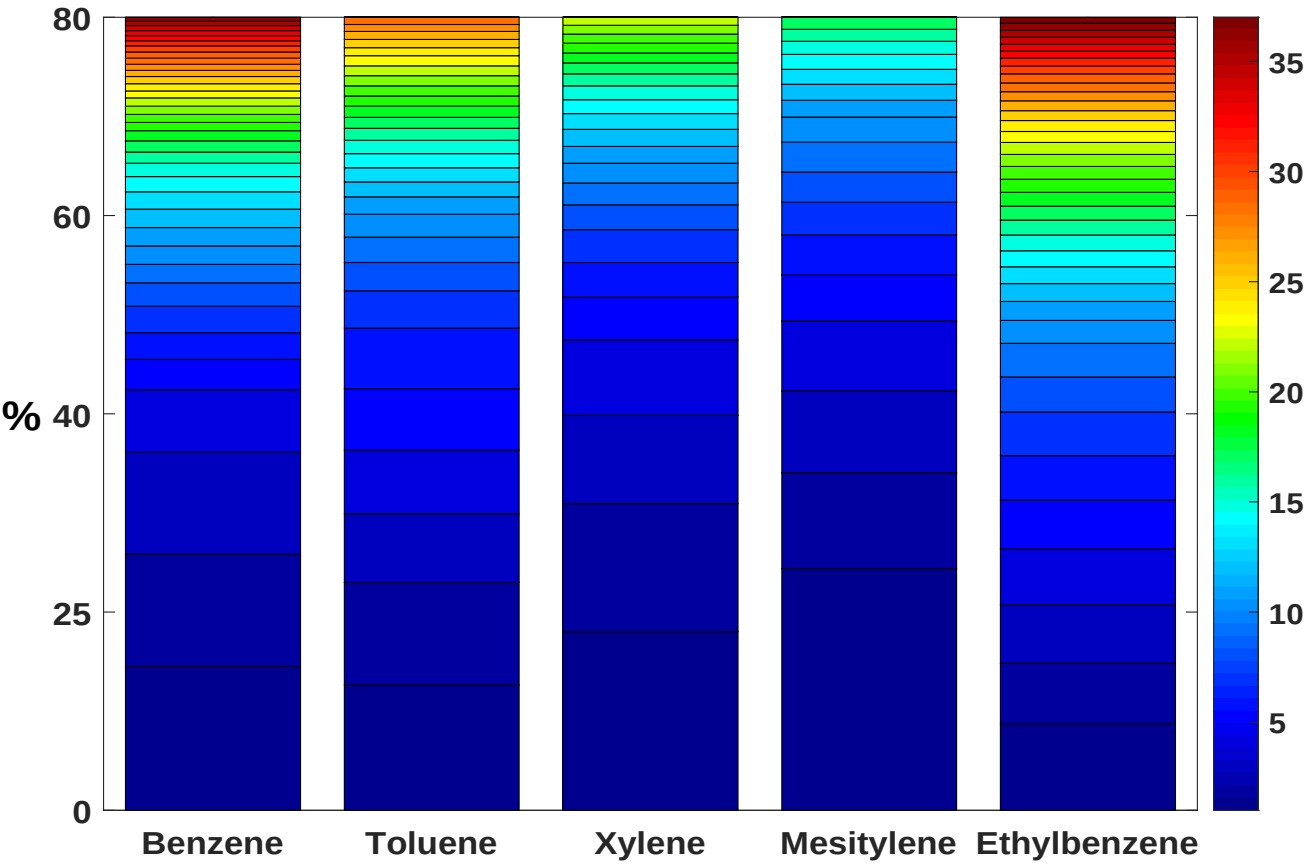

Figure 4. Graphical representation of the cumulative peak lists for the 5 monocyclic ArHC. The bar plot shows the cumulative contribution of the most abundant HOMs species to 80% of the total detected signal for each single-ring ArHC. Each colour of the stacked bar plots denotes a certain number of cumulative compounds. In the series benzene, toluene, xylene and mesitylene a gradually smaller number of different HOM species is needed to explain 80% of the total signal. Ethylbenzene does not follow this empirical observation and shows the highest number of HOMs in these 80%. This may be linked to the fact that dimers with an unexpectedly low number of hydrogen were observed. Ethylbenzene shows a lower fraction of HOM·$NO_3^-$ adducts as well.



Figure 5 Proposed generalized reaction scheme of HOM formation for mesitylene (1,3,5-trimethylbenzene) after OH addition. In black closed-shell species (even number of H), in red radicals (odd number of H). Grey colour denotes radicals that were not detected by the CIMS. Radicals in the orange box are from the propagation of the initial OH attack with an odd number of oxygen, radicals in the yellow box are first generation products with an even number of oxygen, while radicals in the purple boxes are products of a secondary OH addition. Reacting oxygen molecules along the radical propagation chain are not indicated. Closed-shell species are divided into monomers (green boxes) and dimers (blue boxes). The percentages in the boxes indicate the relative intensity of a peak to the total detected ArHC HOMs signal. The sum does not add up to 100% because some peaks mainly coming from fragmentation are not included in the scheme.





Figure 6. Proposed radical reaction mechanism for the autoxidation of mesitylene. The mechanism is derived from the MCM
(version 3.3.1) and takes into account 1[st] generation radicals. The mechanism is intended to represent just one possibility;
other paths are reasonable as well but not shown here.



**Appendix A**

Peak lists of the most abundant HOMs of each ArHC tested are presented in the next 7 tables. The peaks are sorted with decreasing relative intensity. Chemical formula, exact mass and fraction of the explained signal are included. The exact mass includes the mass of $NO_3$ if present.

**Table A-1**

CI-APi-TOF peak list for benzene ($C_6H_6$) oxidation products.

| Chemical formula | Mass | Fraction of explained signal |
|---|---|---|
| $C_6H_8O_5(NO_3)^-$ | 222.0255 | 14.1 |
| $C_5H_6O_8(NO_3)^-$ | 255.9946 | 11.0 |
| $C_6H_6O_2(NO_3)^-$ | 172.0251 | 10.0 |
| $C_{12}H_{14}O_8(NO_3)^-$ | 348.0572 | 6.1 |
| $C_5H_6O_7(NO_3)^-$ | 239.9997 | 3.0 |
| $C_6H_8O_9(NO_3)^-$ | 286.0052 | 2.6 |
| $C_5H_6O_6(NO_3)^-$ | 224.0048 | 2.6 |
| $C_6H_8O_6(NO_3)^-$ | 238.0205 | 2.3 |
| $C_6H_8O_8(NO_3)^-$ | 270.0103 | 1.8 |
| $C_{12}H_{14}O_{14}(NO_3)^-$ | 444.0267 | 1.8 |
| $C_6H_8O_{10}(NO_3)^-$ | 302.0001 | 1.8 |
| $C_6H_6O_4(NO_3)^-$ | 204.0150 | 1.8 |
| $C_6H_8O_7(NO_3)^-$ | 254.0154 | 1.7 |
| $C_6H_{10}O_6(NO_3)^-$ | 240.0361 | 1.5 |
| $C_{12}H_{14}O_{12}(NO_3)^-$ | 412.0369 | 1.3 |
| $C_6H_7O_9(NO_3)^-$ | 284.9974 | 1.1 |
| $C_6H_7O_4^-$ | 143.0350 | 1.1 |
| $C_{12}H_{16}O_9(NO_3)^-$ | 366.0678 | 1.0 |
| $C_6H_8O_4(NO_3)^-$ | 206.0306 | 0.8 |
| $C_6H_6O_5(NO_3)^-$ | 220.0099 | 0.8 |
| $C_{12}H_{14}O_9(NO_3)^-$ | 364.0522 | 0.8 |
| $C_6H_6O_6(NO_3)^-$ | 236.0048 | 0.8 |
| $C_5H_6O_2(NO_3)^-$ | 160.0251 | 0.7 |
| $C_{12}H_{14}O_{10}(NO_3)^-$ | 380.0471 | 0.7 |
| $C_6H_5O_5^-$ | 157.0143 | 0.7 |
| $C_{12}H_{16}O_8(NO_3)^-$ | 350.0729 | 0.6 |
| $C_6H_5O_6^-$ | 173.0092 | 0.6 |
| $C_5H_8O_7(NO_3)^-$ | 242.0154 | 0.6 |
| $C_6H_{10}O_8(NO_3)^-$ | 272.0259 | 0.6 |
| $C_5H_6O_5(NO_3)^-$ | 208.0099 | 0.6 |
| $C_{12}H_{14}O_{11}(NO_3)^-$ | 396.0420 | 0.5 |
| $C_{12}H_{12}O_{13}(NO3)^-$ | 426.0161 | 0.5 |
| $C_6H_7O_6^-$ | 175.0248 | 0.5 |
| $C_{10}H_{10}O_{18}(NO_3)^-$ | 479.9751 | 0.5 |



| | | |
|---|---|---|
| $C_6H_7O_4(NO_3)^-$ | 205.0228 | 0.4 |
| $C_6H_6O_7(NO_3)^-$ | 251.9997 | 0.4 |





**Table A-2**

CI-APi-TOF peak list for toluene ($C_7H_8$) oxidation products.

| Chemical formula | Mass | Fraction of explained signal |
|---|---|---|
| $C_7H_{10}O_9(NO_3)^-$ | 300.0208 | 12.4 |
| $C_{14}H_{18}O_{12}(NO_3)^-$ | 440.0682 | 10.2 |
| $C_7H_{10}O_5(NO_3)^-$ | 236.0412 | 6.8 |
| $C_7H_{10}O_7(NO_3)^-$ | 268.0310 | 6.3 |
| $C_{14}H_{18}O_8(NO_3)^-$ | 376.0885 | 6.1 |
| $C_7H_{10}O_8(NO_3)^-$ | 284.0259 | 6.0 |
| $C_7H_{10}O_6(NO_3)^-$ | 252.0361 | 3.7 |
| $C_7H_8O_2(NO_3)^-$ | 186.0408 | 2.8 |
| $C_{14}H_{18}O_{10}(NO_3)^-$ | 408.0784 | 2.5 |
| $C_7H_{12}O_8(NO_3)^-$ | 286.0416 | 2.3 |
| $C_{14}H_{18}O_9(NO_3)^-$ | 392.0834 | 1.7 |
| $C_6H_7O_6(NO_3)^-$ | 237.0126 | 1.5 |
| $C_7H_{12}O_7(NO_3)^-$ | 270.0467 | 1.4 |
| $C_{14}H_{18}O_{11}(NO_3)^-$ | 424.0733 | 1.4 |
| $C_7H_9O_9(NO_3)^-$ | 299.0130 | 1.3 |
| $C_7H_{10}O_{11}(NO_3)^-$ | 332.0107 | 1.2 |
| $C_7H_9O_{11}(NO_3)^-$ | 331.0029 | 1.1 |
| $C_6H_6O_5^-$ | 158.0215 | 1.1 |
| $C_{14}H_{20}O_{11}(NO_3)^-$ | 426.0889 | 1.1 |
| $C_{14}H_{20}O_9(NO_3)^-$ | 394.0991 | 1.0 |
| $C_7H_8O_4(NO_3)^-$ | 218.0306 | 1.0 |
| $C_6H_8O_8(NO_3)^-$ | 270.0103 | 1.0 |
| $C_{14}H_{18}O_{14}(NO_3)^-$ | 472.0580 | 1.0 |
| $C_7H_{10}O_{10}(NO_3)^-$ | 316.0158 | 1.0 |
| $C_7H_{12}O_6(NO_3)^-$ | 254.0518 | 0.8 |
| $C_7H_8O_5(NO_3)^-$ | 234.0255 | 0.8 |
| $C_7H_8O_6(NO_3)^-$ | 250.0205 | 0.8 |
| $C_{14}H_{20}O_{10}(NO_3)^-$ | 410.0940 | 0.7 |
| $C_7H_8O_8(NO_3)^-$ | 282.0103 | 0.7 |




**Table A-3**

CI-APi-TOF peak list for ethylbenzene ($C_8H_{10}$) oxidation products.

| Chemical formula | Mass | Fraction of explained signal |
|---|---|---|
| $C_8H_{12}O_7(NO_3)^-$ | 282.0467 | 8.7 |
| $C_8H_{12}O_5(NO_3)^-$ | 250.0568 | 6.1 |
| $C_{16}H_{22}O_8(NO_3)^-$ | 404.1198 | 5.9 |
| $C_{16}H_{22}O_{12}(NO_3)^-$ | 468.0995 | 5.6 |
| $C_{16}H_{22}O_{10}(NO_3)^-$ | 436.1096 | 4.9 |
| $C_8H_{12}O_9(NO_3)^-$ | 314.0365 | 4.5 |
| $C_8H_{12}O_6(NO_3)^-$ | 266.0518 | 4.4 |
| $C_8H_9O_8^-$ | 233.0303 | 3.5 |
| $C_8H_{12}O_8(NO_3)^-$ | 298.0416 | 3.4 |
| $C_8H_9O_6^-$ | 201.0405 | 2.3 |
| $C_8H_{10}O_6(NO_3)^-$ | 264.0361 | 1.9 |
| $C_8H_{10}O_{10}(NO_3)^-$ | 328.0158 | 1.8 |
| $C_8H_{14}O_7(NO_3)^-$ | 284.0623 | 1.7 |
| $C_{16}H_{22}O_9(NO_3)^-$ | 420.1147 | 1.6 |
| $C_8H_{10}O_8(NO_3)^-$ | 296.0259 | 1.6 |
| $C_8H_{14}O_8(NO_3)^-$ | 300.0572 | 1.5 |
| $C_{16}H_{20}O_9(NO_3)^-$ | 418.0991 | 1.4 |
| $C_{16}H_{22}O_{14}(NO_3)^-$ | 500.0893 | 1.4 |
| $C_{16}H_{22}O_{11}(NO_3)^-$ | 452.1046 | 1.3 |
| $C_8H_{10}O_2(NO_3)^-$ | 200.0564 | 1.3 |
| $C_{16}H_{24}O_9(NO_3)^-$ | 422.1304 | 1.2 |
| $C_8H_9O_4^-$ | 169.0506 | 1.2 |
| $C_8H_{10}O_9(NO_3)^-$ | 312.0208 | 1.1 |
| $C_8H_{10}O_7(NO_3)^-$ | 280.0310 | 1.1 |
| $C_8H_{12}O_{11}(NO_3)^-$ | 346.0263 | 1.0 |
| $C_8H_{12}O_{10}(NO_3)^-$ | 330.0314 | 1.0 |
| $C_8H_{11}O_6^-$ | 203.0561 | 0.9 |
| $C_{16}H_{24}O_{11}(NO_3)^-$ | 454.1202 | 0.9 |
| $C_7H_8O_5^-$ | 172.0377 | 0.9 |
| $C_8H_9O_3^-$ | 153.0557 | 0.8 |
| $C_8H_{14}O_6(NO_3)^-$ | 268.0674 | 0.8 |
| $C_8H_8O_5^-$ | 184.0377 | 0.7 |
| $C_{16}H_{24}O_{10}(NO_3)^-$ | 438.1253 | 0.7 |
| $C_8H_{10}O_4(NO_3)^-$ | 232.0463 | 0.7 |
| $C_7H_8O_5(NO_3)^-$ | 234.0255 | 0.7 |
| $C_8H_{11}O_9(NO_3)^-$ | 313.0287 | 0.7 |
| $C_7H_7O_4^-$ | 155.0350 | 0.6 |


**Table A-4**

CI-APi-TOF peak list for xylene ($C_8H_{10}$) oxidation products.

| Chemical formula | Mass | Fraction of explained signal |
|---|---|---|
| $C_{16}H_{22}O_8(NO_3)^-$ | 404.1198 | 18.0 |
| $C_8H_{12}O_5(NO_3)^-$ | 250.0568 | 12.9 |
| $C_8H_{12}O_6(NO_3)^-$ | 266.0518 | 8.9 |
| $C_8H_{12}O_7(NO_3)^-$ | 282.0467 | 7.6 |
| $C_{16}H_{22}O_{10}(NO_3)^-$ | 436.1096 | 4.3 |
| $C_8H_{14}O_8(NO_3)^-$ | 300.0572 | 3.5 |
| $C_8H_{14}O_7(NO_3)^-$ | 284.0623 | 3.3 |
| $C_{16}H_{22}O_9(NO_3)^-$ | 420.1147 | 2.5 |
| $C_8H_{12}O_8(NO_3)^-$ | 298.0416 | 2.2 |
| $C_{16}H_{22}O_{12}(NO_3)^-$ | 468.0995 | 2.0 |
| $C_8H_9O_4^-$ | 169.0506 | 1.7 |
| $C_8H_{12}O_9(NO_3)^-$ | 314.0365 | 1.7 |
| $C_8H_{14}O_6(NO_3)^-$ | 268.0674 | 1.6 |
| $C_{16}H_{24}O_{11}(NO_3)^-$ | 454.1202 | 1.4 |
| $C_8H_{11}O_6(NO_3)^-$ | 265.0439 | 1.4 |
| $C_{16}H_{22}O_{11}(NO_3)^-$ | 452.1046 | 1.2 |
| $C_8H_{10}O_6(NO_3)^-$ | 264.0361 | 1.1 |
| $C_8H_{10}O_8(NO_3)^-$ | 296.0259 | 1.0 |
| $C_{16}H_{24}O_{10}(NO_3)^-$ | 438.1253 | 1.0 |
| $C_8H_{10}O_5(NO_3)^-$ | 248.0412 | 0.9 |
| $C_8H_{13}O_8(NO_3)^-$ | 299.0494 | 0.9 |
| $C_8H_{10}O_7(NO_3)^-$ | 280.0310 | 0.8 |





**Table A-5**

CI-APi-TOF peak list for mesitylene (C$_9$H$_{12}$) oxidation products.

| Chemical formula | Mass | Fraction of explained signal |
|---|---|---|
| C$_{18}$H$_{26}$O$_8$(NO$_3$)$^-$ | 432.1511 | 24.2 |
| C$_9$H$_{14}$O$_7$(NO$_3$)$^-$ | 296.0623 | 9.6 |
| C$_9$H$_{14}$O$_6$(NO$_3$)$^-$ | 280.0674 | 8.2 |
| C$_{18}$H$_{26}$O$_{10}$(NO$_3$)$^-$ | 464.1410 | 7.0 |
| C$_9$H$_{16}$O$_7$(NO$_3$)$^-$ | 298.0780 | 4.6 |
| C$_9$H$_{14}$O$_5$(NO$_3$)$^-$ | 264.0725 | 4.0 |
| C$_9$H$_{16}$O$_8$(NO$_3$)$^-$ | 314.0729 | 3.3 |
| C$_9$H$_{16}$O$_9$(NO$_3$)$^-$ | 330.0679 | 3.0 |
| C$_9$H$_{13}$O$_7$(NO$_3$)$^-$ | 295.0545 | 3.0 |
| C$_9$H$_{13}$O$_6^-$ | 217.0718 | 2.5 |
| C$_9$H$_{15}$O$_8$(NO$_3$)$^-$ | 313.0651 | 1.7 |
| C$_{18}$H$_{26}$O$_9$(NO$_3$)$^-$ | 448.1461 | 1.6 |
| C$_9$H$_{14}$O$_8$(NO$_3$)$^-$ | 312.0572 | 1.5 |
| C$_9$H$_{12}$O$_6$(NO$_3$)$^-$ | 278.0518 | 1.5 |
| C$_9$H$_{17}$O$_{10}$(NO$_3$)$^-$ | 347.0705 | 1.3 |
| C$_9$H$_{14}$O$_{10}^-$ | 282.0592 | 1.2 |
| C$_9$H$_{17}$O$_9$(NO$_3$)$^-$ | 331.0756 | 1.2 |



**Table A-6**

CI-APi-TOF peak list for naphthalene ($C_{10}H_8$) oxidation products.

| Chemical formula | Mass | Fraction of explained signal |
|---|---|---|
| $C_{20}H_{18}O_4(NO_3)^-$ | 384.1089 | 26.1 |
| $C_{20}H_{18}O_6(NO_3)^-$ | 416.0987 | 17.4 |
| $C_{20}H_{18}O_5(NO_3)^-$ | 400.1038 | 4.1 |
| $C_{10}H_7O_5^-$ | 207.0299 | 3.5 |
| $C_{20}H_{18}O_7(NO_3)^-$ | 432.0936 | 3.0 |
| $C_{20}H_{18}O_9(NO_3)^-$ | 464.0834 | 2.9 |
| $C_{10}H_{10}O_4(NO_3)^-$ | 256.0463 | 2.5 |
| $C_{10}H_{10}O_5(NO_3)^-$ | 272.0412 | 2.1 |
| $C_{10}H_7O_3^-$ | 175.0401 | 1.8 |
| $C_{10}H_{10}O_8(NO_3)^-$ | 320.0259 | 1.7 |
| $C_{20}H_{18}O_8(NO_3)^-$ | 448.0885 | 1.6 |
| $C_{10}H_{10}O_6(NO_3)^-$ | 288.0361 | 1.5 |
| $C_{10}H_8O_7(NO_3)^-$ | 302.0154 | 1.2 |
| $C_{20}H_{20}O_7(NO_3)^-$ | 434.1093 | 1.1 |
| $C_{10}H_7O_4^-$ | 191.0350 | 1.0 |
| $C_{20}H_{18}O_{10}(NO_3)^-$ | 480.0784 | 1.0 |
| $C_{20}H_{18}O_{11}(NO_3)^-$ | 496.0733 | 1.0 |
| $C_{10}H_{12}O_6(NO_3)^-$ | 290.0518 | 0.9 |
| $C_{10}H_{10}O_7(NO_3)^-$ | 304.0310 | 0.9 |
| $C_{10}H_7O_6^-$ | 223.0248 | 0.9 |
| $C_{10}H_7O_7^-$ | 239.0197 | 0.9 |
| $C_{10}H_8O_5(NO_3)^-$ | 270.0255 | 0.9 |
| $C_{10}H_{10}O_{10}(NO_3)^-$ | 352.0158 | 0.8 |
| $C_{10}H_{10}O_9(NO_3)^-$ | 336.0208 | 0.6 |



**Table A-7**

CI-APi-TOF peak list for biphenyl ($C_{12}H_{10}$) oxidation products.

| Chemical formula | Mass | Fraction of explained signal |
|---|---|---|
| $C_{24}H_{22}O_8(NO_3)^-$ | 500.1198 | 9.1 |
| $C_{12}H_{12}O_5(NO_3)^-$ | 298.0569 | 6.2 |
| $C_{12}H_{14}O_6(NO_3)^-$ | 316.0674 | 5.6 |
| $C_{24}H_{22}O_7(NO_3)^-$ | 484.1249 | 4.9 |
| $C_{24}H_{24}O_7(NO_3)^-$ | 486.1406 | 3.7 |
| $C_{12}H_9O_4^-$ | 217.0506 | 3.6 |
| $C_{12}H_9O_3^-$ | 201.0557 | 3.5 |
| $C_{24}H_{22}O_6(NO_3)^-$ | 468.1300 | 3.5 |
| $C_{24}H_{22}O_9(NO_3)^-$ | 516.1147 | 3.2 |
| $C_{24}H_{24}O_9(NO_3)^-$ | 518.1304 | 2.9 |
| $C_{24}H_{24}O_8(NO_3)^-$ | 502.1355 | 2.8 |
| $C_{24}H_{22}O_{11}(NO_3)^-$ | 548.1046 | 2.5 |
| $C_{12}H_{12}O_6(NO_3)^-$ | 314.0518 | 2.4 |
| $C_{12}H_{12}O_8(NO_3)^-$ | 346.0416 | 2.3 |
| $C_{12}H_{14}O_5(NO_3)^-$ | 300.0725 | 1.7 |
| $C_{12}H_{14}O_7(NO_3)^-$ | 332.0623 | 1.7 |
| $C_{12}H_{12}O_7(NO_3)^-$ | 330.0467 | 1.4 |
| $C_{24}H_{22}O_{10}(NO_3)^-$ | 532.1097 | 1.4 |
| $C_{12}H_{12}O_9(NO_3)^-$ | 362.0365 | 1.4 |
| $C_{11}H_9O_3^-$ | 189.0557 | 1.4 |
| $C_{12}H_9O_5^-$ | 233.0455 | 1.2 |
| $C_{12}H_{10}O_4^-$ | 218.0585 | 1.1 |
| $C_{24}H_{24}O_{10}(NO_3)^-$ | 534.1253 | 1.1 |
| $C_{24}H_{24}O_5(NO_3)^-$ | 454.1507 | 1.1 |
| $C_{24}H_{22}O_{12}(NO_3)^-$ | 564.0995 | 1.1 |
| $C_{24}H_{21}O_8^-$ | 437.1242 | 1.0 |
| $C_{12}H_{12}O_4(NO_3)^-$ | 282.0619 | 1.0 |
| $C_{12}H_{11}O_4^-$ | 219.0663 | 1.0 |
| $C_{10}H_9O_2^-$ | 161.0608 | 0.8 |
| $C_{12}H_{10}O_2(NO_3)^-$ | 248.0564 | 0.8 |
| $C_{36}H_{34}O_{10}(NO_3)^-$ | 688.2036 | 0.8 |
| $C_{12}H_{11}O_5^-$ | 235.0612 | 0.7 |
| $C_{12}H_{10}O_4(NO_3)^-$ | 280.0463 | 0.7 |
| $C_{11}H_7O_2^-$ | 171.0452 | 0.7 |
| $C_{24}H_{22}O_5(NO_3)^-$ | 452.1351 | 0.6 |
| $C_{10}H_7O_3^-$ | 175.0401 | 0.6 |
| $C_{10}H_7O_4^-$ | 191.0350 | 0.6 |




**Appendix B**

HOMs from the 7 tested compounds are presented in the next figures. Top panel left: pie chart showing the monomer and dimer fraction; 3 bar plots presenting the relative signal intensities for radicals, monomer and dimers. In the bar plots, the x
axis presents the number of oxygen atoms and the colour code the number of hydrogen atoms for each of the HOMs.





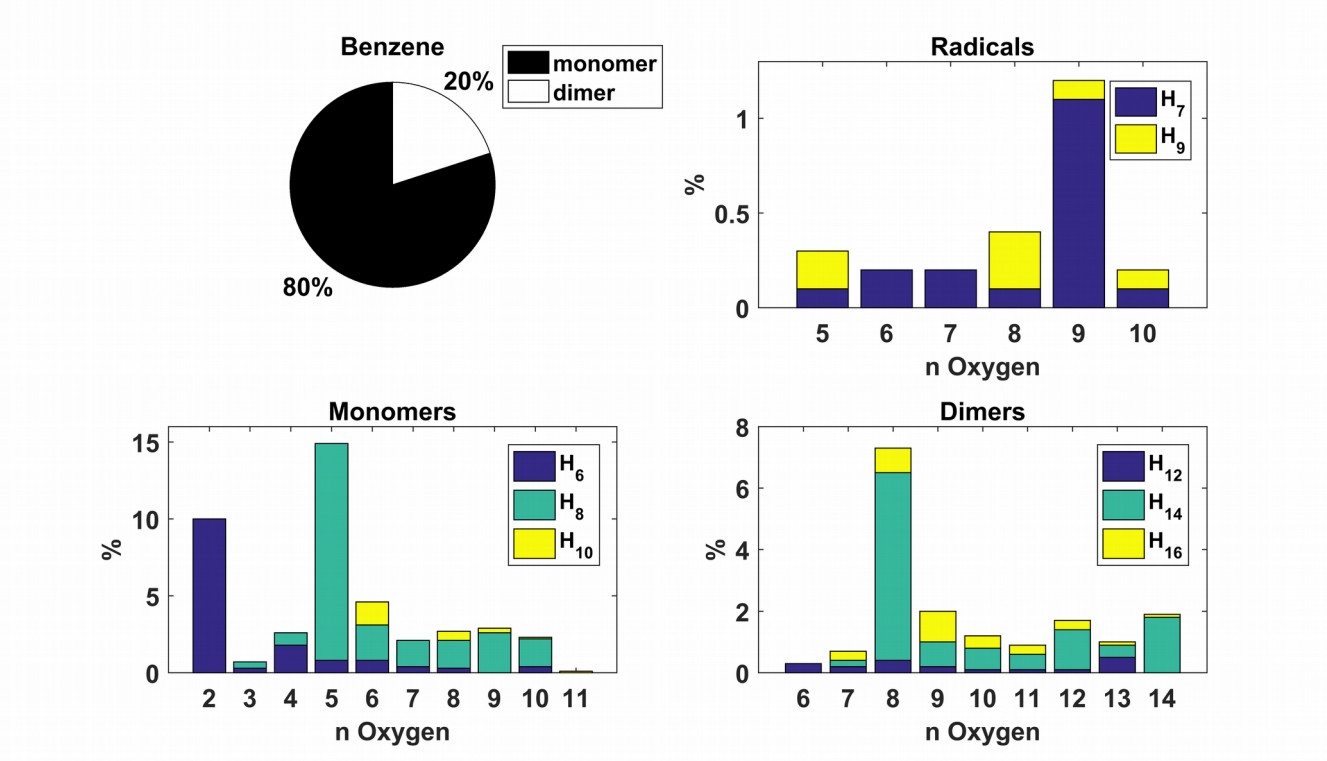

Figure B – 1 Benzene




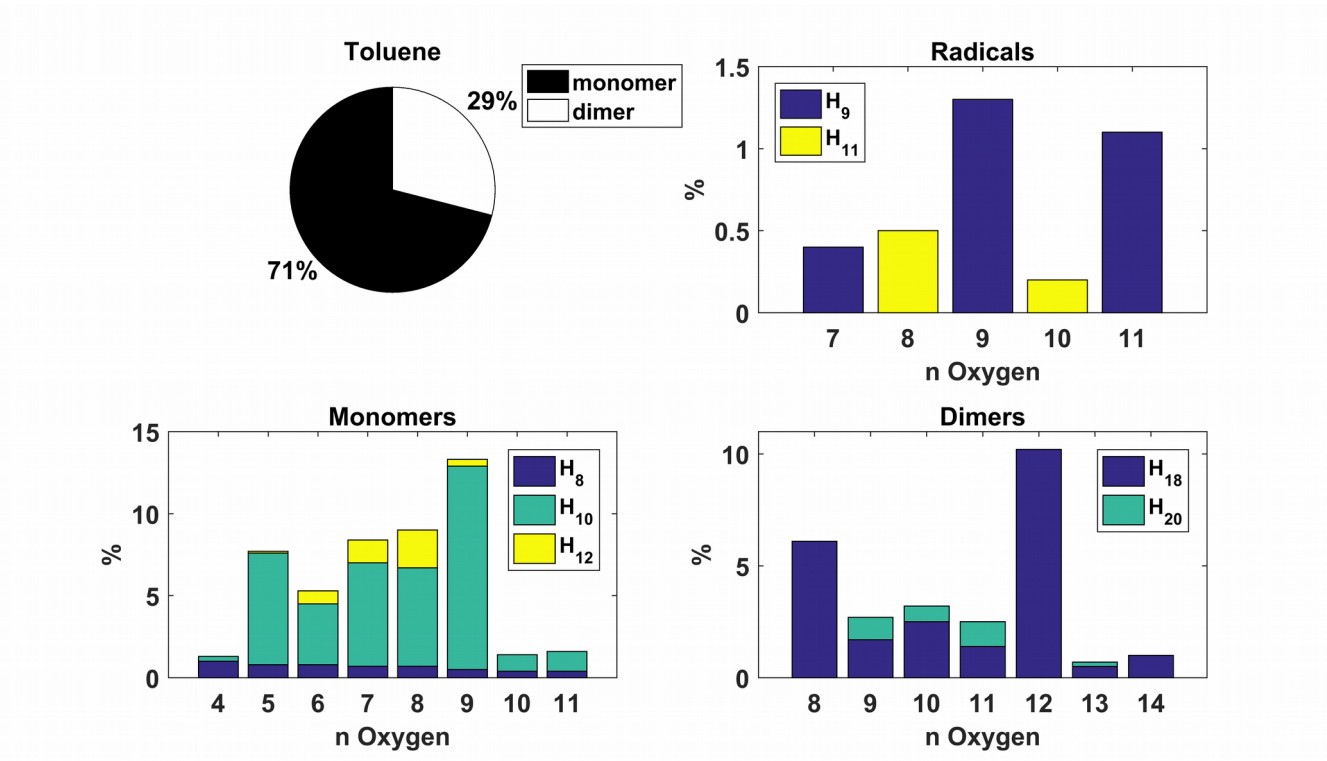

Figure B – 2 Toluene





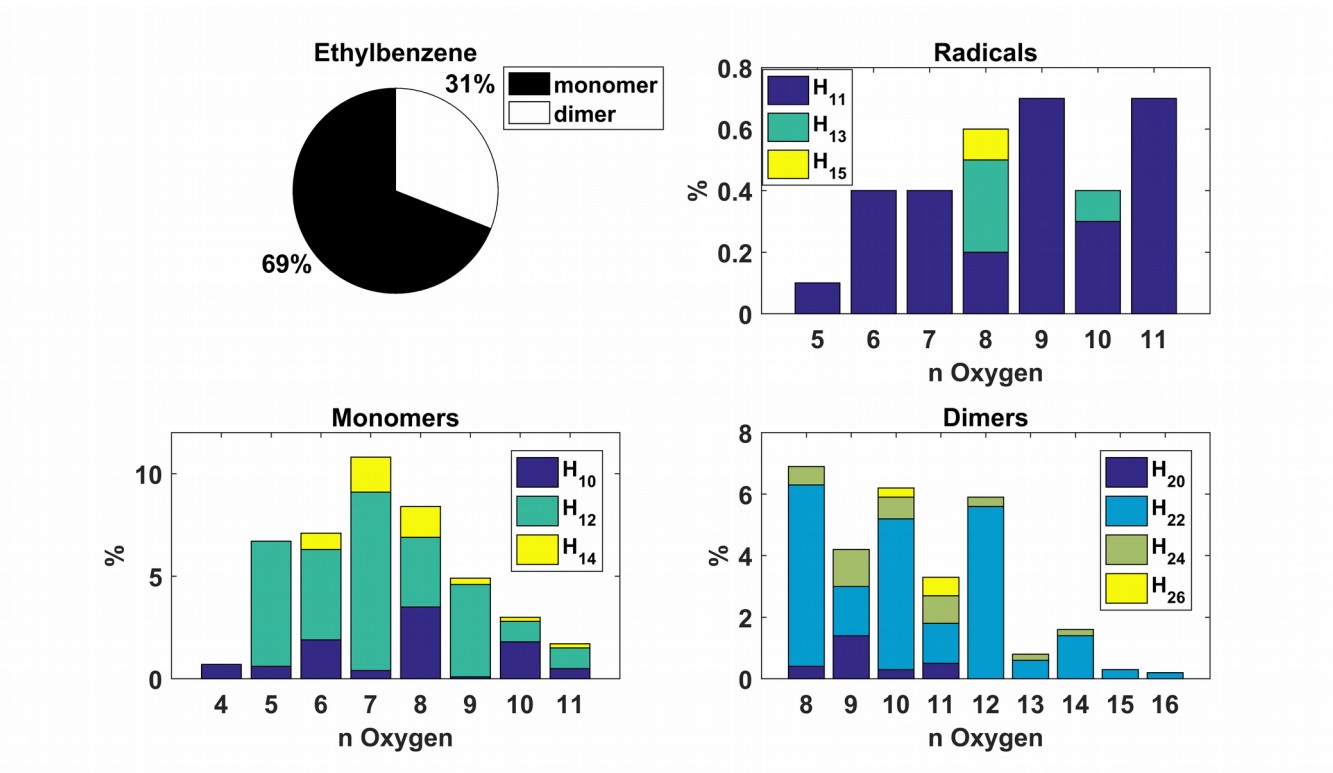

Figure B – 3 Ethylbenzene





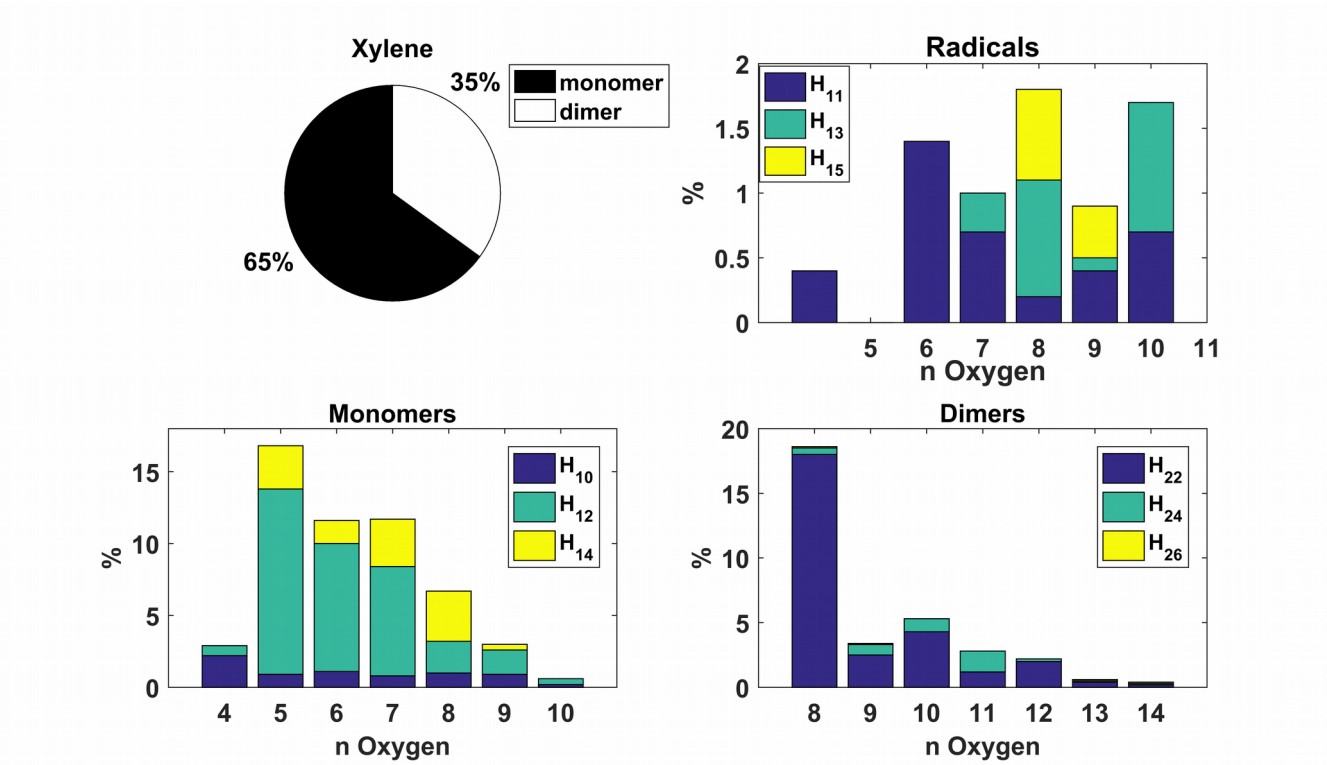

Figure B – 4 Xylene


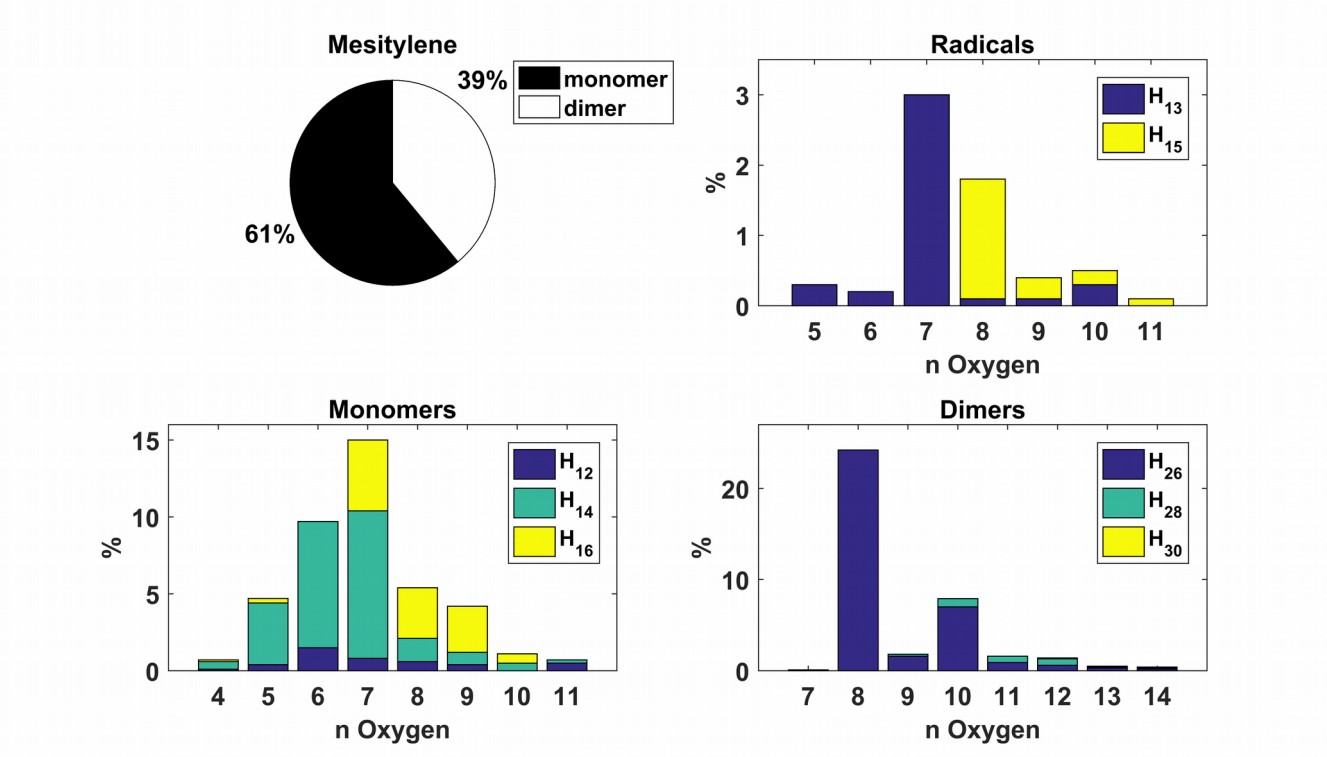

Figure B – 5  Mesitylene





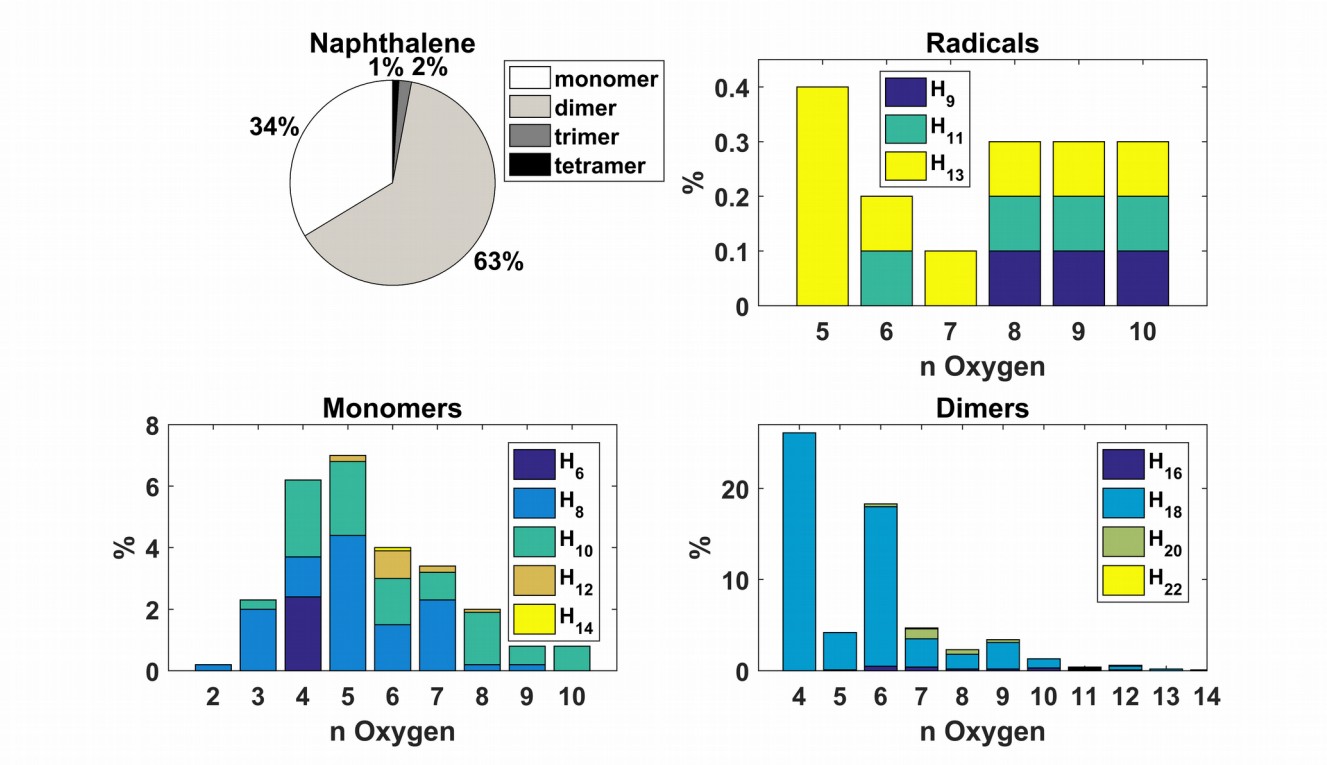

Figure B – 6 Naphthalene






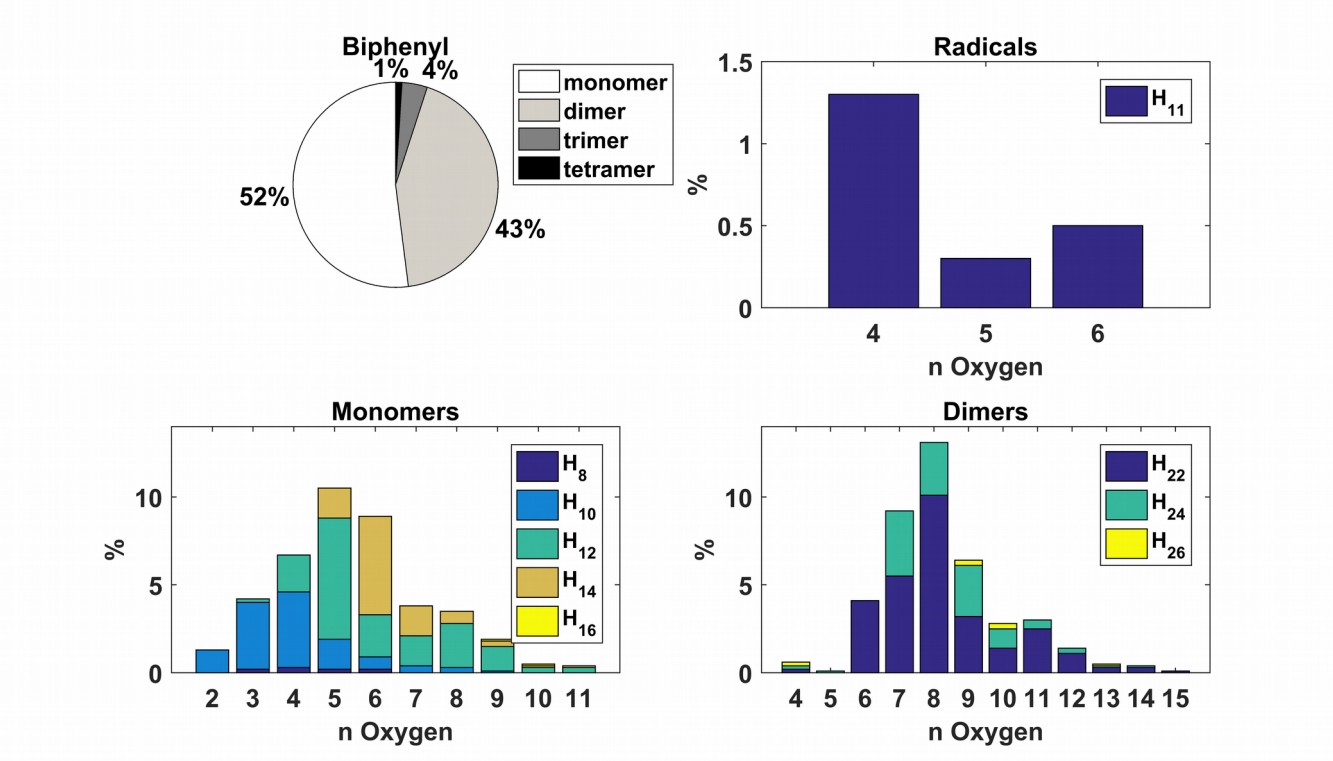

Figure B – 7 Biphenyl