# Peer review of "Formation of highly oxygenated organic molecules from aromatic compounds."

_Atmospheric Chemistry and Physics, 2016_

## Referee Comment (RC1) · Anonymous Referee #1 · 6 Jan 2017

Ms.no. acp-2016-1126

The authors describe experimental findings from a flow-tube study on HOM formation from the reaction of OH radicals with a series of aromatic compounds. OH radicals were generated by water VUV photolysis at 172 nm. The OH+aromatic reaction carried out in a flow system was separated from the OH radical production. Nitrate CIMS was chosen for HOM detection. OH radical concentrations in the system were obtained using an indirect way via a scavenger method. HOM formation from aromatic compounds represents an interesting topic within the framework of SOA precursor formation. It could be a very important process for SOA formation in urban areas. This manuscript needs a couple of clarifications and further explanations before publication can be recommended. Here my comments:

[Figure]

- Line 17: From my perspective, the authors do not present "identified products", they simply show product signals and they discus possible moieties or structural elements of these products.

- Line 48: It is better to say: " . . . aromaticity is lost (or is abrogated)"

- Line 65: OH radicals were generated by water photolysis using a Xe excimer lamp. The atmospheric chemistry community does not commonly use this approach of OH formation. It should be explained here more in detail what are the main reaction steps in a water/air mixture after irradiation at 172 nm. I guess it is also important to comment on the formation of other products like HO2, H2O2 and ozone and their concentration levels in the reaction gas. How important is especially the reaction of OH radicals with HO2 before entering the mixing zone with the flow containing the aromatic compound? It is not enough to state some references only.

- Line 81: Obviously, the reagent ion spectrum showed a strong signal of trifluoroacetate arising from fluorinated contaminants of the Nafion membrane. Was there also a strong signal of the "dimer", i.e. the trifluoroacetate adduct with trifluoroacetic acid? It would be fine to see a reagent ion spectrum as recorded for commonly used reaction conditions (maybe in Appendix). Was the trifluoroacetate concentration low enough that a possible contribution in the ionization process can be excluded?

- Line 84: What was the procedure applied for the determination of the "average" OH radical concentration? Was the disappearance of deuterated butanol monitored by PTR-MS when OH formation was switched on? What was the initial butanol concentration? I guess it has been done in absence of the aromatics, or not? The authors used a kind of a discharge technique. Consequently, the initial OH concentration was much higher than the "average" concentration of 1.9x10(8) molec./cc.? Please provide more information. In addition, the authors could show a figure with OH, aromatic and total HOM concentrations as a function of time or reactor length.

- Line 92: At this point it is not clear where the (molar?) HOM formation yields are

coming from? Did the authors measure the disappearance of the aromatics by PTR-MS, if measureable? Or did the authors calculate the amount of converted aromatics based on their measured OH radical concentration? The concentrations of converted aromatics along with the total HOM concentrations should be stated in Table 1 or in a separate table. What was the calibration factor used for the HOM concentrations measured by nitrate CIMS, where does it come from and what is the detection limit for these HOMs? Please clarify the way of concentration determination for reacted aromatics as well as for the HOMs.

- Line 95 and Fig.2: Main peaks in Fig.2 should be numbered and this numbering should be also given in the corresponding signal table in Appendix. It makes it easier to understand the signal assignment.

- Line 110 and Table 2: The detected product distribution is strongly dependent on the reaction conditions and, therefore, only valid for the conditions of this experiment. That should be clearly mentioned at this point.

- Line 124: What does it mean "more oxygenated radicals have a higher probability to undergo an auto-termination radical reaction"? Please state this pathway and give more information for that.

- Line 161: Do the authors believe that they are able to detect RO radicals by nitrate CIMS? What is the expected RO lifetime with respect to isomerization and for a possible reaction with O2 (depending on the structure)?

- Line 167: "uptake an oxygen atom" ?

- Line 223: At the end of this paragraph a discussion on possible ozone reactions of products is welcome. The ozone concentration in this experiment is quite high and the products have to contain double bonds after losing the aromaticity. Moreover, a statement is needed how the results of this study can be used to explain the formation of SOA precursors for urban conditions where the reaction of RO2+NO dominates the

[Figure]

RO2 fate in most cases.

- Fig.5: What does "Product with DB" mean?

[Figure]

---

## Referee Comment (RC2) · Anonymous Referee #2 · 24 Jan 2017

Review of: Formation of highly oxygenated organic molecules from aromatic compounds. By Molteni, U. et al.

Significance This is an interesting work describing the important observation of highly-oxidized molecules (HOM) from aromatic oxidation reactions. Although certainly important observation in a current "hot-topic" field, and as such should merit its publication, I have a problem how the results are presented. It almost seems as this has been written with a format more suitable for general wider audience in a magazine format, and not really addressed to the atmospheric chemist and physics community. I feel that a certain amount of details of the experiments and the setup have been omitted, which could greatly help researchers in the field performing these type of experiments. Due to this formatting issue in many cases it seems that the text just assumes too much from the reader. This will become clearer from the large amount of specific comments

given below and starting with: "What do you mean by..".

So while I find the topic extremely interesting and the finding important, I cannot recommend publication in the current form. I further stress that most of this is due to the current presentation form and all of the problems can be fixed relatively easily. Thus I strongly suggest that the authors take time to modify/rewrite the text according to the comments given below, after which I can recommend publishing.

Major Comments:

Abstract should contain the major details of the work described in the manuscript, i.e., what was studied with what methods and what were the main results, and Conclusions should contain summary of the results and their significance. At the moment they do not do this. More precisely, currently Abstract is missing many of these details and Conclusions feels more like an extension of Discussion. I suggest significantly improving the current presentation. In the Abstract you talk about identified compounds, although there were no single compound really identified in the whole study. This should be changed. In the Abstract you talk about mechanistic pathways – was there more than the one shown for mesitylene in Figure 6? Also in line 53 you say that you talk about "potential pathways and possible mechanism", but I feel that currently the mechanistic aspects are only briefly discussed. More details of the OH radical production, especially the geometry and the distance OH needs to travel, could be helpful. It was stated that this setup has been used to generate $HO_2$, which is co-produced by $H_2O$ photolysis in presence of $O_2$, but $HO_2$ is much less reactive and thus can travel much further, whereas OH is easily lost to impurities and walls (and as far as I understood OH needs to travel through two 90 degree bends, from which at least the other is a turbulent zone?). Was there only one experiment with one hydrocarbon? Was the OH production always exactly the same? These should be stated clearly. It would be very helpful to give the precursor structures to help the reader understand the significance of these oxidation processes. Line 84: Could you include a figure showing the process of OH concentration determination as it is very central to the whole topic. Line 86: You

talk about reactive non-aromatic double bonds together with 140 ppbv ozone – how is it "expected" that O3 reactions do not play a role here? Line 112: Would you expect the dimer/monomer ratio to be constant? If, then why so? This should be stated. Furthermore, what do you mean by this being a good proxy? How and why? Line 117: How much of the given abundances could be actually due to shifting transmission of the mass spec? Is this a potential problem? Line 118: I guess this also assumes that the k(RO2 + HO2) is similar for all systems? Should be stated. Moreover, according to Table 1 different VOC concentrations where used (which have different rate coefficients with OH), which leads to different [RO2] and to different strength of RO2 + RO2, right? Line 127: You should clearly separate speculation, i.e., do you know if the stated radicals are forming observed dimers? Lines 145 to 174: In talking about products which main difference is the amount of O-atoms, it seems a bit confusing that all of them have the same label "z". Could you think of any other way of representing them so that it would not seem they all have the same "z-amount" of O-atoms. Line 174: Does this formula mean that you found products with 4 and 6 H-atoms more than in the parent VOC?. How could you get to a product with 6 H-atoms more than in the parent structure? Can you give an example how this could happen? Line 181: What do you mean by "lower than expected H-atom number"? Line 192: What do you mean by "much less H-atoms than terpenes"? Line 192: Methyl group is not generally considered a good leaving group. Could you add a brief explanation or a reference? Line 194: The occurrence of multiple OH attacks seems somewhat obvious from the observed product compositions. However, the given OH concentration together with such a short residence time in the flow tube does not seem to allow much 2nd generation oxidation (and of course even less 3rd generation). I think this fact should be addressed in the text. Perhaps a chemical reaction simulation could help to get an idea of the needed RO2 lifetimes and the OH + product reaction rates to justify the high amounts of products evident from figures. Actually, the figures seem to indicate way higher product concentrations than what was the used reagent concentration (=[OH]). For example, Figure 3 gives a concentration of >3x109 cm-3 for a single product, even though the stated

[OH] was only 2x108 cm-3 which should equal to the maximum product concentration, right? Line 201: What do you mean by "conjugated radical in an allylic position"? This should be rewritten. Line 205: What do you mean by "This mechanism varies among the ArHC tested."? How is the mechanism changing? Line 211: A reference should be added for: "termination reactions of alkoxy radicals make double bonds". Is that a possible reaction for alkoxy radicals? Line 229: Oxygen bridged bicyclic radicals were not discussed in the text. Line 230: The oxygen addition to allylic positions was not really discussed in the text. Line 236: There was no discussion about phenolic structures in the main text. As already stated above, the current conclusions would rather fit as a continuation of discussion and not as a "conclusions" chapter. Line 245: Please rewrite sentence starting with "Furthermore..". Figure 1: Excimer. Figure 2: I wonder what was your calibration procedure for the mass spectrometer? Figure 4 caption: What do you mean by "ethylbenzene does not follow this empirical observation"? What is the empirical observation, and how ethylbenzene does not follow it? Figure 5: This figure needs some improvement: How do you go from C9H13O3 through alkoxy radical to C9H13O6? Also, you say that yellow box contains even oxygen species, even though it has also odd oxygen species. Table A-1: You talk about species loosing methyl groups in the text, but what explains the benzene products with less C-atoms than the parent?

Minor Comments:

Line 35: I don't think CCN (anymore) exert an influence on pre-industrial times (i.e., be careful with the wording). Line 44: I guess it's really hard to prove that something does not happen, right? So I'm a bit wondering why so many references have been grouped to indicate that no HOMs were seen by studies that did not use the current methods able to detect the HOMs in the first place. Line 49: Multiple non-aromatic double bonds exists also in many terpenoid species, so it's wrong to argue that this is a property of aromatics alone. Line 67: I think it could help the reader if you could provide a bit more details about OH generation. At least I cannot fully understand the method how it's described now. Do you have an injector or why is the coaxial geometry mentioned? Line

79: "acid-base reaction" – do you mean salt formation? Line 96: The stated "monomer" and "dimer" ranges overlap. Line 99: Which one is 14 Th – methyl or ethyl group...? Line 107: What "a mechanism for Van der Waals interactions.." means? Is there a "mechanism"? Line 108: I don't understand why would you talk about 800 Th cluster as a particle while talking about mass spectrometric results? Could you explain the significance of adding this here? Line 121: I wonder if a "monocyclic dimer" is a correct term here, if you do not know if the ring is retained in the reaction or not. Maybe better to reword to state that it is a "dimer" that was generated by monocyclic ArHC. Line 125: What is an "auto-termination reaction"? Line 125: Where is the comparison between unimolecular and bimolecular channels based on? Line 128: What is a "higher-order cluster"? Line 134: add -s to "pathway". Line 150: Both products have same number of O-atoms (=z). Line 153: Rather odd number of H-atoms? Line 157: Is Hyttinen 2015 the right reference for this? Line 162: I don't think Kirkby 2016 is generally a good reference for peroxy radical mechanisms as it seems to contain all the mechanistic aspects in its supplementary material. Line 166: Also reaction 4b forms RO radicals. Line 167: Oxygen molecule. Line 169: What do you mean by "discrepancy between the intensity of the peaks"? Line 196: For which compounds the third OH attack is seen? Line 211: Should be Kurten 2015? Line 218: Too less C atoms in biphenyl dimer. Line 219: Aromatic rings are generally considered rather unreactive than reactive. Line 234: Should be "and" not "or". Line 467: Should be "due" not "doe". Table 1 caption: Why do you state "mixing ratio" here? Table 2: Would it make more sense to express the fraction in percentage so that the O:C and the fractional part would not be mixed so easily? Figure 3 caption: There are no compositions in the given inserts, although they're mentioned. Can the mass spec really retrieve accurate compositions for the pentamers? Line 242: I'm not sure if you should talk about identification here, rather "have curiously the same compositions as..". In addition I think also this part should be in Discussion section. Appendix B: Figures of the parent compounds also here would make the figures more interesting.

---

## Referee Comment (RC3) · Anonymous Referee #3 · 27 Jan 2017

General

The authors describe a flow tube study of formation of highly oxidized molecules from aromatic precursors. The material presented is new and interesting. The paper is interesting to read and the data a presented in a suited manor. The paper contains some degree of speculations about the proposed mechanism but still tolerable as it may induce more research and discussions. After considering the minor comments below the paper should be published in ACP.

Minor comments

p.4, l.92: If you present molar yields, you must have information about the sensitivity of your mass spectrometer. In addition the y-axis in Figures 2 and 3 seem to be given in molecule concentration. (If not, that should be clarified in the captions.) At

other parts of the manuscript you mention that you cannot quantify dimers because the transmission of your TOF-MS is unknown. In this manuscript, any information about the sensitivity of your instrument is missing. How did you estimate the molar yields then? Please, state precisely in the experimental section, what you did to determine the sensitivity or what the basis of your assumptions is.

p4, l.119: Does the split off of O2 explain the lower oxidation degrees of dimers assuming RO2+RO2 = ROOR +O2 or not?

p.4, l.119f and p.6, l.162: I would propose to give here Mentel et al. 2015 somewhat more recognition as they described, based on experimental observations in context of autoxidation, this type of dimer formation including mixed dimers a year before Kirkby et al. 2016. The same is true for the alkoxy path.

p.4, l.192ff: "Additionally, more oxygenated radicals have a higher probability to undergo an auto-termination radical reaction compared to a radical-radical recombination (RO2· + RO2· or RO2· + HO2·)." I don't exactly what you want to say with this statement in context of degree of methylation and dimer fraction. Less methylated aromatic compounds tend to more auto-termination? What means auto-termination - termination by internal reaction?

p.8, l231ff: I think, that one should differentiate clearer between autoxidation by H-shift to peroxy radicals on one hand and by attack of the peroxy moiety to internal double bonds on the other hand. Although both reactions are internal rearrangements they are still of different character, as the first needs "mobile" H-atoms and the latter double bonds with potential to allyl radical formation. As a consequence the HOM formation in aromatic systems would be based - at least in parts- on a different mechanism?!

p.10, l.284: I urgently request to give a full, correct bibliographic reference to the book by Calvert. You can find it, if you google it is a little bit ridiculous.

Typos

p.4, l.96 Thomson

p.7, l.218; "strongest dimer is C12H14O8 for benzene and C12H22O8 for biphenyl, respectively". I guess a typo, as C12H22O8 cannot be a dimer resulting from biphenyl.
* * *

---

## Author Comment (AC1) · 30 Oct 2017

**Author's response:**

We thank the Referee for the careful revision and comments, which helped improving the overall quality of the manuscript.
A point-by-point answer to the referee's remarks is detailed in the following (in black the referee comments, in blue our answers, in green text modifications)

The authors describe experimental findings from a flow-tube study on HOM formation from the reaction of OH radicals with a series of aromatic compounds. OH radicals were generated by water VUV photolysis at 172 nm. The OH+aromatic reaction carried out in a flow system was separated from the OH radical production. Nitrate CIMS was chosen for HOM detection. OH radical concentrations in the system were obtained using an indirect way via a scavenger method. HOM formation from aromatic compounds represents an interesting topic within the framework of SOA precursor formation. It could be a very important process for SOA formation in urban areas. This manuscript needs a couple of clarifications and further explanations before publication can be recommended. Here my comments:

Line 17: From my perspective, the authors do not present "identified products", they simply show product signals and they discuss possible moieties or structural elements of these products.
We agree that not the chemical structures but only the molecular formulas of the HOMs were identified. We modified the text accordingly:
We report the elemental composition of the HOMs and show the differences in the oxidation patterns of these ArHCs.

Line 48: It is better to say: "aromaticity is lost (or is abrogated)"
We agree and we modified this:
When the aromaticity is lost by the OH addition, non-aromatic double bonds are …

Line 65: OH radicals were generated by water photolysis using a Xe excimer lamp. The atmospheric chemistry community does not commonly use this approach of OH formation. It should be explained here more in detail what are the main reaction steps in a water/air mixture after irradiation at 172 nm. I guess it is also important to comment on the formation of other products like HO2, H2O2 and ozone and their concentration levels in the reaction gas. How important is especially the reaction of OH radicals with HO2 before entering the mixing zone with the flow containing the aromatic compound? It is not enough to state some references only.
We include in the appendix section (Appendix C) a description of the OH radical formation with the Xe excimer lamp and a model of the chemistry in these experiments.
**Appendix C**
The radiation at 172 nm excites molecular oxygen and water vapor triggering the following radical reactions:

| | | | |
|---|---|---|---|
| $hv + O_2$ | → | $O(^1D) + O(^3P)$ | (R-C1) |
| $O(^1D) + M$ | → | $O(^3P) + M$ | (R-C2) |
| $O(^3P) + O_2 + M$ | → | $O_3 + M$ | (R-C3) |
| $O(^1D) + H_2O$ | → | $2\ OH$ | (R-C4) |
| $hv + H_2O$ | → | $H + OH$ | (R-C5) |
| $H + O_2 + M$ | → | $HO_2 + M$ | (R-C6) |
| $OH + O_3$ | → | $HO_2 + O_2$ | (R-C7) |
| $HO_2 + O_3$ | → | $OH + 2O_2$ | (R-C8) |
| $OH + HO_2$ | → | $H_2O + O_2$ | (R-C9) |
| $HO_2 + HO_2$ | → | $H_2O_2 + O_2$ | (R-C10) |
| $H_2O_2 + OH$ | → | $H_2O + HO_2$ | (R-C12) |

The humidified air flow is exposed to the 172 nm radiation for 50 ms and is then within 30 ms transferred to the mixing zone with the sample flow. The oxidant species produced are OH, $HO_2$, $O_3$ and $H_2O_2$. The final OH concentration entering the mixing zone depends on the residence time in the lamp and in the transfer region.

The kinetic model includes 31 species and 36 reactions from the MCM 3.3.1 (Jenkin et al., 2003). Mesitylene is selected as ArHC for these simulations and its reaction mechanism is extended up to the second generation products. The model is run for 20 seconds in agreement with the residence time of the flow tube reactor with a simulation time resolution of 2 ms. The model is initiated with the measured concentrations of ozone ($3.45\ 10^{12}$ molecules $cm^{-3}$ without mesitylene) and mesitylene ($2.46\ 10^{12}$ molecules $cm^{-3}$ without lamp on) at the exit of the flow tube. The initial OH radical concentration ($8.50\ 10^{11}$ molecules $cm^{-3}$) is tuned in order to match the OH exposure, which was determined from the amount of reacted mesitylene. The initial $HO_2$ radical concentration ($1.70\ 10^{12}$ molecules $cm^{-3}$) is set at twice the initial OH radical concentration. Wall losses of about 35% are estimated for mesitylene HOMs but are not implemented in the model. Figure C1 shows the temporal evolution of 6 selected species: reacted mesitylene (T135MB reacted), $HO_2$ radical (HO2), OH radical (OH) as well as 3 products of the mesitylene oxidation with the OH radical (TM135BPRO2, TM135BPOOH and TM135B2OH). TM135BPRO2 is an intermediate peroxy radical after OH attack; TM135BPOOH is a product from the reaction of TM135BPRO2 with the $HO_2$ radical while TM135BPO2OH is a product from the reaction of TM135BPRO2 with a peroxy radical $RO_2$. Mesitylene reacted reaches a plateau after about 0.03 seconds while TM135BPRO2 reaches a maximum value around 0.01-0.02 seconds and then rapidly decreases. The closed shell products TM135BPOOH and TM135BPO2OH constantly increase and reach a plateau after about 0.4-1.0 seconds. A similar trend could be expected for HOMs assuming that TM135BPRO2 undergoes an autoxidation chain and is terminated either by $HO_2$ or by $RO_2$. In a test run where the initial mesitylene concentration and the reaction rate constant towards OH radicals were doubled, the ratio TM135BPOOH/TM135BPO2OH varied only by about 18%.

[Figure]

Line 81: Obviously, the reagent ion spectrum showed a strong signal of trifluoroacetate arising from fluorinated contaminants of the Nafion membrane. Was there also a strong signal of the "dimer", i.e. the trifluoroacetate adduct with trifluoroacetic acid? It would be fine to see a reagent ion spectrum as recorded for commonly used reaction conditions (maybe in Appendix). Was the trifluoroacetate concentration low enough that a possible contribution in the ionization process can be excluded?

The main source of trifluoroacetate is indeed the Nafion membrane. Additional experiments (not presented here) with a bubbler showed a much lower trifluoroacetate signal. The trifluoroacetate monomer was the main peak followed by the adduct with nitric acid and the trifluoroacetate dimer. In the experiments where the trifluoroacetate signal was rather high we did not observe adducts with HOMs. From this we conclude that trifluoroacetate did not influence the ionization of the analyte.

Line 84: What was the procedure applied for the determination of the "average" OH radical concentration? Was the disappearance of deuterated butanol monitored by PTR-MS when OH formation was switched on? What was the initial butanol concentration? I guess it has been done in absence of the aromatics, or not? The authors used a kind of a discharge technique. Consequently, the initial OH concentration was much higher than the "average" concentration of 1.9x10(8) molec./cc.? Please provide more information. In addition, the authors could show a figure with OH, aromatic and total HOM concentrations as a function of time or reactor length

The OH concentration was determined from the difference in precursor concentration between excimer lamp switched on or off. Two D9-butanol, toluene, mesitylene, and biphenyl experiments resulted in an average value of $2 \cdot 10^8$ OH radicals cm$^{-3}$. It was not possible to derive an OH

concentration from the benzene, ethylbenzene and naphthalene experiments due to technical difficulties. As xylene was a mixture of 3 isomers with different OH-reaction rate constants, the OH concentration could not be determined. We assume that the OH production was constant in all experiments as the lamp was operated under similar conditions. We developed a kinetic model to simulate the decrease of the aromatic compounds under the conditions of our flow tube experiments. This is described in the appendix (Appendix section C), which includes a figure with the concentrations of OH, aromatic precursor and a HOM representative as a function of time (see description above).

Line 92: At this point it is not clear where the (molar?) HOM formation yields are coming from? Did the authors measure the disappearance of the aromatics by PTRMS, if measureable? Or did the authors calculate the amount of converted aromatics based on their measured OH radical concentration? The concentrations of converted aromatics along with the total HOM concentrations should be stated in Table 1 or in a separate table. What was the calibration factor used for the HOM concentrations measured by nitrate CIMS, where does it come from and what is the detection limit for these HOMs? Please clarify the way of concentration determination for reacted aromatics as well as for the HOMs.

The HOM molar yield was calculated from the ratio of HOM concentration to reacted ArHC (the yields were not corrected for the HOMs losses). The amount of ArHC reacted was calculated based on the assumption, that the OH production was similar in all experiments (see comment above). The HOMs quantification is based on the calibration factor for sulfuric acid and the assumption that HOMs have the same ionization efficiency as sulfuric acid (Ehn et al., 2014; Kirkby et al., 2016). It is known that the ionization efficiency depends on the structure of the HOMs. Since the ionization efficiency might be <1 for some HOM structures, the reported concentrations here represent lower limits. We added the following in the text:

HOMs yields are calculated as the ratio of HOMs measured to ArHC reacted. HOMs were quantified using the calibration factor for sulfuric acid and assuming the same charging efficiency for HOMs (Ehn et al., 2014; Kirkby et al., 2016). From the decrease of the precursor concentration (lights off versus lights on) we determined an average OH concentration. From the experiments with D9-butanol, toluene, mesitylene and biphenyl an average OH concentration of $2 \times 10^8$ OH cm$^{-3}$ was obtained. Assuming the same OH production of the lamp in all experiments ArHC reacted was calculated from the OH radical exposure using the reaction rate coefficients at 25°C.

Table 1 was complemented with HOMs concentration, reacted fraction of ArHCs and HOMs yield and looks now like this:

Table 1

Initial concentrations of precursors, reaction rate coefficients, ArHC reacted fraction (%), total HOMs concentration and HOMs yield (%). The mixing ratio of precursors was determined at the exit of the flow tube when the excimer lamp (OH generation) was switched off.

| Compound | Concentration (molecules cm$^{-3}$) | $k_{OH}$ ($10^{-12}$ cm$^3$ molecules$^{-1}$ s$^{-1}$) | Reacted fraction (%) | [HOM] (molecules cm$^{-3}$) | HOMs yield (%) |
|---|---|---|---|---|---|
| Benzene ($C_6H_6$) | $9.85 \times 10^{13}$ | 1.22 | 0.5 | $1.2 \times 10^9$ | 0.2 |
| Toluene ($C_7H_8$) | $1.97 \times 10^{13}$ | 5.63 | 2.3 | $4.4 \times 10^8$ | 0.1 |
| Ethylbenzene ($C_8H_{10}$) | $1.13 \times 10^{13}$ | 7.0 | 2.8 | $9.4 \times 10^8$ | 0.3 |

| | | | | | |
|---|---|---|---|---|---|
| (o/m/p)-xylene (C$_8$H$_{10}$) | 2.95 10$^{12}$ | 13.6/23.1/14.3 | // | 2.8 10$^9$ | // |
| Mesitylene (C$_9$H$_{12}$) | 2.46 10$^{12}$ | 56.7 | 22.7 | 3.1 10$^9$ | 0.6 |
| Naphthalene (C$_{10}$H$_8$) | 2.95 10$^{13}$ | 23.0 | 9.2 | 1.4 10$^{10}$ | 0.5 |
| Biphenyl (C$_{12}$H$_{10}$) | 4.43 10$^{13}$ | 7.1 | 2.8 | 1.8 10$^{10}$ | 1.4 |

Reference for the *k*-rates: (Atkinson and Arey, 2003)

Line 95 and Fig.2: Main peaks in Fig.2 should be numbered and this numbering should be also given in the corresponding signal table in Appendix. It makes it easier to understand the signal assignment.

We updated Figure 2 by adding for each compound the chemical structure on the left side of the figure. To help the reader we included labels with the oxygen number in the monomer and dimer panels.

See Figure 2.

Line 110 and Table 2: The detected product distribution is strongly dependent on the reaction conditions and, therefore, only valid for the conditions of this experiment. That should be clearly mentioned at this point.

We agree with this. The new text reads:

However, since the mass-dependent ion transmission efficiency is rather smooth the given values may faithfully represent the relative product distribution of the different aromatic compounds.

Line 124: What does it mean "more oxygenated radicals have a higher probability to undergo an auto-termination radical reaction"? Please state this pathway and give more information for that.

This statement can be derived from (Rissanen et al., 2014): We rewrote it as follows:

Additionally, more oxygenated radicals have a higher probability to undergo a unimolecular termination compared to a radical-radical recombination (RO$_2$· + RO$_2$· or RO$_2$· + HO$_2$·). More oxygen atoms imply more peroxy functional groups and therefore a higher probability of a hydrogen abstraction in geminal position of a peroxide which results in an OH radical loss and a carbonyl group formation.

Line 161: Do the authors believe that they are able to detect RO radicals by nitrate CIMS? What is the expected RO lifetime with respect to isomerization and for a possible reaction with O2 (depending on the structure)?

We do not detect alkoxy radicals. Based on our understanding their lifetime is too short to yield detectable concentrations. From the fact that compounds with even numbers of oxygen are observed we infer that the alkoxy radical isomerizes to a carbon centered radical and takes up oxygen molecules in an auto-oxidation mechanism. This is explained in the text in the lines 166-168.

Line 167: "uptake an oxygen atom"?

We replaced atom with molecule.

can again take up an oxygen molecule.

Line 223: At the end of this paragraph a discussion on possible ozone reactions of products is welcome. The ozone concentration in this experiment is quite high and the products have to

contain double bonds after losing the aromaticity. Moreover, a statement is needed how the results of this study can be used to explain the formation of SOA precursors for urban conditions where the reaction of RO2+NO dominates the RO2 fate in most cases.

While the ArHC do not react with ozone, oxidation products with remaining double bonds indeed may. The ozone concentration produced in the Xe excimer lamp is about 140 ppb. The residence time in the flow tube is 20 s. Reaction rate constants for the ozonolysis of alkenes are in the range of $10^{-16} - 10^{-18}$ cm$^3$ molecule$^{-1}$s$^{-1}$. Under the condition that 20 ppb of ArHC have reacted and yield 10% of products with a double bond still containing the carbon skeleton of the parent molecule, about 0.14 - 14 ppt of these products will react with ozone. Assuming a high HOMs yield of 10% from this ozonolysis reaction we can expect 0.014-1.4 ppt of HOMs from ozonolysis. This can be compared to 60-800 ppt HOMs produced via OH radical attack. Thus, the contribution of ozonolyis to HOMs formation can be neglected.

We added the following to the text:

[revised manuscript text omitted]

---

## Author Comment (AC2) · 30 Oct 2017

**Author's response:**

We thank the Referee for the careful revision and comments, which helped improving the overall quality of the manuscript.
A point-by-point answer to the referee's remarks is detailed in the following (in black the referee comments, in blue our answers, in green text modifications)

Significance This is an interesting work describing the important observation of highly oxidized molecules (HOM) from aromatic oxidation reactions. Although certainly important observation in a current "hot-topic" field, and as such should merit its publication, I have a problem how the results are presented. It almost seems as this has been written with a format more suitable for general wider audience in a magazine format, and not really addressed to the atmospheric chemist and physics community. I feel that a certain amount of details of the experiments and the setup have been omitted, which could greatly help researchers in the field performing these type of experiments. Due to this formatting issue in many cases it seems that the text just assumes too much from the reader. This will become clearer from the large amount of specific comments given below and starting with: "What do you mean by..". So while I find the topic extremely interesting and the finding important, I cannot recommend publication in the current form. I further stress that most of this is due to the current presentation form and all of the problems can be fixed relatively easily. Thus I strongly suggest that the authors take time to modify/rewrite the text according to the comments given below, after which I can recommend publishing.

Major Comments:

Abstract should contain the major details of the work described in the manuscript, i.e., what was studied with what methods and what were the main results, and Conclusions should contain summary of the results and their significance. At the moment they do not do this. More precisely, currently Abstract is missing many of these details and Conclusions feels more like an extension of Discussion. I suggest significantly improving the current presentation.
We have modified Abstract and Conclusions accordingly

In the Abstract you talk about identified compounds, although there were no single compound really identified in the whole study. This should be changed.
Referee #1has raised a similar comment. We agree and modified the sentence to:
We report the molecular formulas of the HOMs and show the differences in the oxidation patterns of these ArHCs.

In the Abstract you talk about mechanistic pathways – was there more than the one shown for mesitylene in Figure 6?
We show a possible oxidation pathway for mesitylene as an example case. The other single ring aromatics may follow a similar scheme, although branching ratios may vary. We now say:
A potential pathway for the formation of these HOMs from aromatics is presented and discussed.

Also in line 53 you say that you talk about "potential pathways and possible mechanism", but I feel that currently the mechanistic aspects are only briefly discussed.
Indeed, the lack of unambiguous identification of molecular structures does not allow for an extensive evaluation of the mechanism. We present a generalized scheme of a mechanism, which represents the observed elemental composition of species, and discuss a potential pathway of HOMs formation. We changed the wording such that it should be clear that we do not provide proofs of a detailed mechanism. The sentence reads now:
Here we show the formation of HOMs from ArHC upon reaction with OH radicals. We present product distributions of HOMs in terms of molecular masses and molecular formulas for a series of

aromatic precursors based on measurements with a nitrate chemical ionization atmospheric pressure interface time of flight mass spectrometer (CI-APi-TOF) (Ehn et al., 2014; Jokinen et al., 2012; Kürten et al., 2011). A potential pathway along with a possible mechanism for the formation of HOMs from aromatic compounds is discussed.

More details of the OH radical production, especially the geometry and the distance OH needs to travel, could be helpful. It was stated that this setup has been used to generate HO2, which is co-produced by H2O photolysis in presence of O2, but HO2 is much less reactive and thus can travel much further, whereas OH is easily lost to impurities and walls (and as far as I understood OH needs to travel through two 90 degree bends, from which at least the other is a turbulent zone?). Was there only one experiment with one hydrocarbon?

The humidified air flow is exposed to the 172 nm radiation for 50 ms and is then within 30 ms transferred to the mixing zone with the sample flow. The oxidant species entering the mixing zone are OH, $HO_2$, $O_3$ and $H_2O_2$. Their concentrations depend on the residence time in the lamp and the reaction time in the transfer region. This is now explained in Appendix C together with the chemical reactions forming the oxidants. Despite potentially higher losses of OH to the walls compared to $HO_2$, there was still enough OH present to react with the precursor gases. The same setup was used for all precursors and for D9-butanol. From this we derived an initial OH concentration of about 8.5 $10^{11}$ cm$^{-3}$.
See Appendix C.

Was the OH production always exactly the same? These should be stated clearly.

The OH concentration was determined from the difference in precursor concentration between excimer lamp switched on or off. Two D9-butanol, toluene, mesitylene, and biphenyl experiments resulted in an average value of 2 $10^8$ OH radicals cm$^{-3}$. It was not possible to derive an OH concentration from the benzene, ethylbenzene and naphthalene experiments due to technical difficulties. As xylene was a mixture of 3 isomers with different OH-reaction rate constants, the OH concentration could not be determined. We assume that the OH production was constant in all experiments as the lamp was operated under similar conditions.

From the decrease of the precursor concentration (lights off versus lights on) we determined an average OH concentration. From the experiments with D9-butanol, toluene, mesitylene and biphenyl an average OH concentration of 2 $10^8$ OH cm$^{-3}$ was obtained. Assuming the same OH production of the lamp in all experiments ArHC reacted was calculated from the OH radical exposure using the reaction rate coefficients at 25°C.

It would be very helpful to give the precursor structures to help the reader understand the significance of these oxidation processes.

The chemical structures of the precursors are now included in Figure 2.

Line 84: Could you include a figure showing the process of OH concentration determination as it is very central to the whole topic.

An average OH concentration was determined from the decrease of the precursor gases with the lamp off and on. Using a kinetic model we derived the initial OH concentration and the profile of the precursors. This is now provided in Appendix C.

Line 86: You talk about reactive non-aromatic double bonds together with 140 ppbv ozone – how is it "expected" that O3 reactions do not play a role here?

This question was also raised by referee #1.

While the ArHC do not react with ozone, oxidation products with remaining double bonds indeed do. The ozone concentration produced in the Xe excimer lamp is about 140 ppb. The residence time in the flow tube is 20 s. Reaction rate constants for the ozonolysis of alkenes are in the range of $10^{-16}$ – $10^{-18}$ cm$^3$ molecule$^{-1}$s$^{-1}$. Under the condition that 20 ppb of ArHC have reacted and yield 10% of

products with a double bond still containing the carbon skeleton of the parent molecule, about 0.14 - 14 ppt of these products will react with ozone. Assuming a high HOMs yield of 10% from this ozonolysis reaction we can expect 0.014-1.4 ppt of HOMs from ozonolysis. This can be compared to 60-800 ppt HOMs produced via OH radical attack. Thus, the contribution of ozonolyis to HOMs formation can be neglected.

Line 112: Would you expect the dimer/monomer ratio to be constant? If, then why so? This should be stated. Furthermore, what do you mean by this being a good proxy? How and why?
We do not expect the dimer to monomer ratio to be the same for all precursors. Indeed the ratios might be somewhat different when measured with different instruments because the transmission functions are not the same. Since the transmission of ions may vary with the voltage settings in the two quadrupoles in the APi, dimer:monomer ratios may differ between instruments. However, since the transmission curve is expected to be rather smooth the relative trend of the dimer/monomer within the 5 single-ring ArHC may still be reasonably accurate due to the small mass differences in the monomer and dimer ranges. We modified the sentence:
However, since the mass-dependent ion transmission efficiency is rather smooth the given values may faithfully represent the relative behavior of the product distribution of the different aromatic compounds.

Line 117: How much of the given abundancies could be actually due to shifting transmission of the mass spec? Is this a potential problem?
As commented above this should have a small effect on the relative changes of the monomer:dimer ratio. This could be a problem if the instrument were tuned in such a way that the ion transmission is strongly mass dependent. Usually, one tries to avoid this.

Line 118: I guess this also assumes that the k(RO2 + HO2) is similar for all systems? Should be stated. Moreover, according to Table 1 different VOC concentrations where used (which have different rate coefficients with OH), which leads to different [RO2] and to different strength of RO2 + RO2, right?
Yes, we assume a similar $k(RO_2 + HO_2)$. We think it is a reasonable assumption that the reaction rate $RO_2 + HO_2$ does vary less with different substituents compared to the reaction of $RO_2$ with $RO_2$. The master chemical mechanism also uses just one rate constant for this reaction.
The lamp produces a similar concentration of OH and $HO_2$ in all experiments independent of the precursor concentration. This means that the concentration of $RO_2$ produced and the amount of $HO_2$ present is similar in the different experiments. In our sensitivity test with the flow tube kinetic model for mesitylene we selected the species TM135BPOOH and TM135BP2OH as proxy for products of $RO_2$-$HO_2$ and $RO_2$-$RO_2$ reactions. When the initial mesitylene concentration and the reaction rate coefficient with the OH radical were doubled the ratio TM135BPOOH / TM135BP2OH changed only by 18%. Therefore we believe that the observed differences among the tested ArHC compounds can be attributed to a different selectivity rather than to a different radical concentration. We changed the text to:
This indicates that the branching ratio of $RO_2 + RO_2$·to dimer (R3c) compared to the other reaction channels (R3a,b) is higher for the more substituted aromatics. This is based on the assumption that the lamp produces similar concentrations of OH and HO2 radicals and that the reaction rate coefficients $k(RO_2 + HO_2)$ (R4a) are similar for all $RO_2$.

Line 127: You should clearly separate speculation, i.e., do you know if the stated radicals are forming observed dimers?
We agree, we replaced the text with:
Furthermore, we assume that less oxygenated radicals, although not quantitatively detected by the CI-APi-TOF (Berndt et al., 2015; Hyttinen et al., 2015), will nevertheless participate in the dimer formation.

Lines 145 to 174: In talking about products which main difference is the amount of O-atoms, it seems a bit confusing that all of them have the same label "z". Could you think of any other way of representing them so that it would not seem they all have the same "z-amount" of O-atoms.

We consider "z" as a variable which can be any number while we focus on the hydrogen atom number. To make it more explicit we added:

Where z denotes any number of oxygen atoms,

Line 174: Does this formula mean that you found products with 4 and 6 H-atoms more than in the parent VOC?. How could you get to a product with 6 H-atoms more than in the parent structure?
Can you give an example how this could happen?

HOMs with 6 hydrogen atoms more were in general a very small fraction of the total detected HOMs signal (this is true for the dimers as well). As an example one could think that the first OH adds one hydrogen atom to the initial ArHC and the termination of the radical chain to an alcohol or hydroperoxy function adds a second hydrogen atom. If the remaining 2 double bonds are not cleaved within the radical chain propagation this mechanism can be repeated up to 3 times, each time adding 2 hydrogen atoms. This can indeed lead to HOMs with 6 hydrogen atoms more than the ArHC precursor. This was already described in lines 194-197.

Line 181: What do you mean by "lower than expected H-atom number"?
We mean less hydrogen atoms than the precursor. We write now:
Similarly, the compounds with an H-atom number lower than the ArHC precursor could have been formed by an H-abstraction from first generation products with formula $C_xH_yO_z$.

Line 192: What do you mean by "much less H-atoms than terpenes"?
We mean that the fragmentation could involve the elimination of a fragment that contains hydrogen atoms instead of CO alone.
HOMs with less C atoms than the parent molecule have also been previously described from terpene precursors via CO elimination (Rissanen et al., 2014, 2015). Here, the aromatics show mostly also a loss of H-atoms when fragmenting. This indicates that a methyl group can be lost after oxidation to an alkoxy radical as formaldehyde or a carbon fragment can be lost after ring cleavage.

Line 192: Methyl group is not generally considered a good leaving group. Could you add a brief explanation or a reference?
If an H-atom is abstracted from a methyl group an alkoxy radical can be formed, which can decompose with the loss of $H_2CO$. See corrected text above.

Line 194: The occurrence of multiple OH attacks seems somewhat obvious from the observed product compositions. However, the given OH concentration together with such a short residence time in the flow tube does not seem to allow much 2nd generation oxidation (and of course even less 3rd generation). I think this fact should be addressed in the text. Perhaps a chemical reaction simulation could help to get an idea of the needed RO2 lifetimes and the OH + product reaction rates to justify the high amounts of products evident from figures. Actually, the figures seem to indicate way higher product concentrations than what was the used reagent concentration (=[OH]). For example, Figure 3 gives a concentration of >3x109 cm-3 for a single product, even though the stated [OH] was only 2x108 cm-3 which should equal to the maximum product concentration, right?

The reported OH radical concentration is an average value. Due to the fast reaction the OH radical concentration is much higher at the beginning of the flow tube. Initial OH levels are about 8.5 $10^{11}$ $cm^{-3}$ and rapidly drop off a few orders of magnitude. Up to 23% of the precursor gas reacts and

therefore an OH attack on the first generation products can occur in such a short reaction time. We report the modeled OH radical concentration in the Appendix C.

Line 201: What do you mean by "conjugated radical in an allylic position"? This should be rewritten.

We rephrased the whole sentence, see reply to comment on line 229 below

Line 205: What do you mean by "This mechanism varies among the ArHC tested."? How is the mechanism changing?

We delete this sentence. Variations in the mechanism are provided just thereafter in the text.

Line 211: A reference should be added for: "termination reactions of alkoxy radicals make double bonds". Is that a possible reaction for alkoxy radicals?

The manuscript is not correctly cited. We say that when the radical chain is interrupted the molecule can still have double bonds which are reactive sites towards OH radical attack.

Line 229: Oxygen bridged bicyclic radicals were not discussed in the text.

This was shown in line 204 and Figure 6. We rephrased it as follows:

After addition of OH and loss of aromaticity an oxygen molecule can be added forming a peroxy radical. It has been established that the latter can cyclize producing a second stabilized allylic radical with an endocyclic $O_2$ bridge (Baltaretu et al., 2009; Birdsall and Elrod, 2011; Pan and Wang, 2014). On this oxygen bridged bicyclic radical further oxygen addition and cyclization might occur up to a peroxy radical with seven oxygen atoms ($C_9H_{13}O_7$), which is the species detected at relatively high intensity (3.5%).

Line 230: The oxygen addition to allylic positions was not really discussed in the text.

It is now mentioned, see above.

Line 236: There was no discussion about phenolic structures in the main text. As already stated above, the current conclusions would rather fit as a continuation of discussion and not as a "conclusions" chapter.

We agree with the referee, we move this part to the results and discussion section and rephrased it:

Recent studies (Nakao et al., 2011; Schwantes et al., 2017) suggest a mechanism where the initial step is the formation of the phenolic equivalent ArHC followed by additional oxidation steps yielding "polyphenolic" structures with high O:C ratio (up to 1.2). However literature data are showing varying yields for the conversion of arenes to phenols via the OH radical addition and H elimination. According to MCM 3.3.1 (Jenkin et al., 2003) benzene and toluene have quite high phenol yields (approximately 50 and 20 %, respectively) while mesitylene shows a rather small yield (4%). This fact should be reflected in the final HOMs yield with alkyl substituted ArHCs being less effective in yielding HOMs. However in our experiments we did not detect such a difference in the HOMs yields linked to phenol formation yields. A relevant fraction of the detected HOMs showed a hydrogen atom number higher than the precursor ArHC which cannot be explained with the presence of just polyphenolic compounds as oxidation products.

Line 245: Please rewrite sentence starting with "Furthermore..".

We replaced it with:

Furthermore, the fact that the oxidation of ArHC can rapidly form HOMs of very low volatility makes ArHC a potential contributor to nucleation and early particle growth during nucleation episodes observed in urban areas.

Figure 1: Excimer.
We corrected eximer with excimer.

Figure 2: I wonder what was your calibration procedure for the mass spectrometer?
The HOMs quantification is based on the calibration factor for sulfuric acid ($6.5 \ 10^9 \ cm^{-3}$) and the assumption that HOMs have the same ionization efficiency as sulfuric acid (Ehn et al., 2014; Kirkby et al., 2016). It is known that the ionization efficiency depends on the structure of the HOMs. Since the ionization efficiency might be <1 for some HOM structures, the reported concentrations here represent lower limits. No further characterization of the system CIMS - ArHC HOMs was pursued.

Figure 4 caption: What do you mean by "ethylbenzene does not follow this empirical observation"? What is the empirical observation, and how ethylbenzene does not follow it?
With empirical observation we want to indicate the fact that with the increase of the number of the substituents less HOMs species are needed to sum up to the 80% of the total HOMs signal. While this is true for the series benzene, toluene, xylene, mesitylene, ethylbenzene deviates from this trend. This text now reads:
However, this trend with the increase of the number of substituents is not met by ethylbenzene

Figure 5: This figure needs some improvement: How do you go from C9H13O3 through alkoxy radical to C9H13O6? Also, you say that yellow box contains even oxygen species, even though it has also odd oxygen species.
The scheme is a simplified representation of the HOMs formation for mesitylene. We made two changes to the Figure: 1) The alkoxy radical chain is now in a separate box and 2) we added C9H13O4 to the alkoxy radicals. The arrow links now the odd-oxygen radical box with the alkoxy (even oxygen number) box, from which then the even-oxygen radicals can be produced. The formation of the alkoxy radical can occur from any odd-oxygen peroxy radical. For example C9H13O6 can be formed via two pathways:
$C_9H_{13}O_3 \rightarrow C_9H_{13}O_5 \rightarrow alkoxy \rightarrow C_9H_{13}O_4 \rightarrow C_9H_{13}O_6$
$C_9H_{13}O_3 \rightarrow alkoxy \rightarrow C_9H_{13}O_2 \rightarrow C_9H_{13}O_4 \rightarrow C_9H_{13}O_6$
We replace line 490 with:
Radicals in the orange box are from the propagation of the initial OH attack with an odd number of oxygen atoms, radicals in the pink box are formed via an alkoxy intermediate step with an even number of oxygen atoms, while radicals in the purple box are products of a second OH addition.

Table A-1: You talk about species loosing methyl groups in the text, but what explains the benzene products with less C-atoms than the parent?
In the text we explained how a methyl group can be oxidized and become a good leaving group (e.g. formaldehyde). This is not possible for benzene. However if the autoxidation leads to a ring opening step the resulting radicals can undergo carbon chain fragmentation yielding small fragments (e.g. carbon monoxide, glyoxal).

Minor Comments:

Line 35: I don't think CCN (anymore) exert an influence on pre-industrial times (i.e., be careful with the wording).
We agree and changed the wording to:
CCN can impact climate via their influence on cloud properties; this changes the radiation balance nowadays and did even more so in the pre-industrial period (Carslaw et al., 2013; Gordon et al., 2016).

Line 44: I guess it's really hard to prove that something does not happen, right? So I'm a bit wondering why so many references have been grouped to indicate that no HOMs were seen by studies that did not use the current methods able to detect the HOMs in the first place.

We rewrote this to read:

Despite the fact that ArHC·OH adducts under atmospheric conditions react with $O_2$ to yield peroxy radicals (Calvert et al., 2002; Glowacki et al., 2009; Suh et al., 2003) it is not known if ArHC oxidation also yields HOMs. No carbon balance could be reached so far and generally only about 50% of the carbon reacted was identified as products (Calvert et al., 2002). When aromaticity is lost by the OH addition non-aromatic double bonds are formed representing highly reactive products to more oxidants, which is a peculiar behavior not observed in other classes of VOC (Calvert et al., 2002).

Line 49: Multiple non-aromatic double bonds exists also in many terpenoid species, so it's wrong to argue that this is a property of aromatics alone.

Here we want to say that if ArHC loose their aromaticity products can also become reactive towards other oxidants like $O_3$ and $NO_3$ (Calvert et al., 2002). We now point this out in the previous answer.

Line 67: I think it could help the reader if you could provide a bit more details about OH generation. At least I cannot fully understand the method how it's described now. Do you have an injector or why is the coaxial geometry mentioned?

More information is now provided in Appendix C. We mention the coaxial geometry because coaxial is the geometry of the excimer lamp we used for this series of experiments.

Line 79: "acid-base reaction" – do you mean salt formation?

We mean the abstraction of an acidic H-atom from the HOM by $NO_3^-$, see reaction R2.

Line 96: The stated "monomer" and "dimer" ranges overlap.

Yes, since we studied ArHc with increasing molar mass it is not possible to give an unequivocal range for monomers and dimers for all species.

Line 99: Which one is 14 Th – methyl or ethyl group. . .?

An H-atom is replaced by a methyl group: Methyl ($CH_3$(15 Th) – H(1 Th)). In case of ethylbenzene a $CH_2$ group is added compared to toluene. We changed the sentence to:

….is shifted by differences of 14 Th ($CH_2$) each from benzene via toluene and xylene/ethylbenzene to mesitylene due to the additional substituent groups

Line 107: What "a mechanism for Van der Waals interactions.." means? Is there a "mechanism"?

We replaced this with:

Most of the higher n-mers are probably bonded by intermolecular interactions, similar to biogenic HOMs (Donahue et al., 2013).

Line 108: I don't understand why would you talk about 800 Th cluster as a particle while talking about mass spectrometric results? Could you explain the significance of adding this here?

We made this link because clusters of this size can already be detected by particle counters. Thus mass spectrometry and particle measurements start to overlap and mass spectrometry can help to identify which compounds are participating in NPF. We modified the sentence:

Clusters with m/z ≥ 800 Th might already be detected by particle counters with a mobility diameter d ≥1.5 nm

Line 121: I wonder if a "monocyclic dimer" is a correct term here, if you do not know if the ring is retained in the reaction or not. Maybe better to reword to state that it is a "dimer" that was generated by monocyclic ArHC.

We agree and replaced this with:

while dimers that were generated from monocyclic ArHC have on average

Line 125: What is an "auto-termination reaction"?

It is an intramolecular decomposition that leads to a non-radical organic species and an OH radical as an example. We assessed a similar question from referee #1, to now read:

Additionally, more oxygenated radicals have a higher probability to undergo a unimolecular termination compared to a radical-radical recombination ($RO_2\cdot$ + $RO_2\cdot$ or $RO_2\cdot$ + $HO_2\cdot$. More oxygen atoms imply more peroxy functional groups and therefore a higher probability of a hydrogen abstraction in geminal position of a peroxide group which results in an OH radical loss and a carbonyl group formation. Therefore, the fraction of dimer formation should decrease with higher oxygen content.

Line 125: Where is the comparison between unimolecular and bimolecular channels based on?

As explained before, more oxygen atoms in the HOM imply more peroxide functional groups which increases the probability of the unimolecular pathway. This decreases the fraction of dimers with increasing oxygen content.

Line 128: What is a "higher-order cluster"?

With higher-order cluster we indicate n-mers where the precursor monomeric structure appears 3 or more times. While in the text we refer to them as trimers, tetramers and pentamers when the monomer structure appears 3, 4 and 5 times, respectively, we want to clarify that if monomers and dimers are produced via radical unimolecular termination or the radical-radical reactions reported in the text (R3a-c and R4a,b) higher order clusters (trimers, tetramers, pentamers) are most likely resulting from the aggregation of these monomer and dimer HOMs via the establishment of van der Walls interactions.

Line 134: add -s to "pathway".

We replaced this with:

The increasing number of methyl groups appears to influence the oxidation pathways and leads to less HOM products.

Line 150: Both products have same number of O-atoms (=z).

We corrected the chemical formulas.

Line 153: Rather odd number of H-atoms?

In case of an OH addition an O-atom is added followed by $O_2$ additions, which leads to an odd number of oxygen atoms (zo). In case of an H-atom abstraction $O_2$ adds to the carbon radical and further autoxidation would then produce an even number of oxygen atoms (ze).

Line 157: Is Hyttinen 2015 the right reference for this?

Yes, it is.

Line 162: I don't think Kirkby 2016 is generally a good reference for peroxy radical mechanisms as it seems to contain all the mechanistic aspects in its supplementary material.

We removed Kirkby 2016 as a reference here.

Line 166: Also reaction 4b forms RO radicals.

We replaced this with:
These alkoxy radicals (R3a, R4b) may isomerize to an alcohol by internal H-abstraction forming a carbon centered radical.

Line 167: Oxygen molecule.
We replaced atom with molecule.

Line 169: What do you mean by "discrepancy between the intensity of the peaks"?
We find a prevalence of monomers with formulae $C_xH_{y+2}O_z$ compared to those with formulae $C_xH_yO_z$. On average reaction R3b should yield similar concentrations. From the flow tube kinetic model in Appendix C we note that the $HO_2$ radical concentration is very high during the whole experiment because it is also formed in the source. From this we infer that the radical termination reaction R4a is the main sink of the peroxyradicals under these specific conditions of our experiments .
We say now:
The much higher intensity of the peaks with formula $C_xH_{y+2}O_z$ compared to those with the composition $C_xH_yO_z$ can be ascribed to a high contribution from the recombination of $RO_2\cdot$ with $HO_2\cdot$(R4a). This is due to the high $HO_2$ concentration in our experiments since $HO_2$ is also formed in the OH radical source.

Line 196: For which compounds the third OH attack is seen?
Benzene, ethylbenzene, xylene, naphthalene and biphenyl show HOMs that can be linked to a third OH attack. The contribution of these HOMs to the total of the detected signals is however always extremely low.
A third OH attack is observed only for some compounds: benzene, ethylbenzene, xylene, naphthalene and biphenyl; the contribution of these HOMs to the total of the detected signals is always extremely low. The mechanism will likely proceed in a similar way.

Line 211: Should be Kurten 2015?
The referee is right, we wanted to refer here to: Kurtén, T., Rissanen, M. P., Mackeprang, K., Thornton, J. A., Hyttinen, N., Jørgensen, S., Ehn, M. and Kjaergaard, H. G.: Computational study of hydrogen shifts and ring-opening mechanisms in α-pinene ozonolysis products, J. Phys. Chem. A, 119(46), 11366–11375, doi:10.1021/acs.jpca.5b08948, 2015.

Line 218: Too less C atoms in biphenyl dimer.
This was indeed a typo. We replaced with:
$C_{24}H_{22}O_8$.

Line 219: Aromatic rings are generally considered rather unreactive than reactive.
We wanted to say that the remaining aromatic ring is still quite reactive towards a second attack by OH radicals. We now say:
Compounds with extra-high H-atoms are more frequently found for biphenyl, which is expected as there is a second reactive aromatic ring remaining after (auto)-oxidation of the first one.

Line 234: Should be "and" not "or".
We agree and changed this to and.

Line 467: Should be "due" not "doe".
done

Table 1 caption: Why do you state "mixing ratio" here?
We replaced it with concentration.

Table 2: Would it make more sense to express the fraction in percentage so that the O:C and the fractional part would not be mixed so easily?

Good suggestion. We express now monomer and dimer fractions in percentage.

Figure 3 caption: There are no compositions in the given inserts, although they're mentioned. Can the mass spec really retrieve accurate compositions for the pentamers?

This class of mass spectrometer shows a resolution of 4-5000 with a flattening above mass 150-200 Th. Such resolution is certainly insufficient alone when it comes to provide a peak chemical composition at high masses. However, during the peak analysis process it is possible to infer the chemical composition of the peaks present in the mass spectra, taking advantage of the following assumptions:

- the elements possibly present are carbon, hydrogen; oxygen and nitrogen as a nitrate ion,
- the ratio between these elements has to allow a reasonable chemical structure (e.g. unsaturation number),
- the precursor carbon chain gives some constraints on the number of carbon atoms expected in the chemical formula with no fragmentation due to the ionization process because of the soft ionization method,
- the peak distribution can suggest a repetition of building blocks (e.g. monomer, dimer, trimer, tetramer) as well as peak compositions that differ by 2 hydrogen atoms (2.0157 amu) or 1 oxygen atom (15.9949 amu),

We do not report the chemical composition in the insert, however, all the peaks plotted here are identified with their chemical composition. Pentamers from biphenyl were identified up to mass 1242.3307 Th corresponding to the chemical formula $C_{60}H_{60}O_{25}(NO_3)^-$.

Line 242: I'm not sure if you should talk about identification here, rather "have curiously the same compositions as..". In addition I think also this part should be in Discussion section.

We agree. We now close the manuscript with a paragraph "Discussion and atmospheric implications" and we replaced the text with:

Some of the HOMs measured here from the oxidation of ArHC have the same composition as the HOMs formulae identified by Bianchi et al. (2016) during winter time nucleation episodes at the Jungfraujoch High Altitude Research Station.

Appendix B: Figures of the parent compounds also here would make the figures more interesting.

We included the ArHC structures in each figure.

References

[revised manuscript text omitted]

---

## Author Comment (AC3) · 30 Oct 2017

**Author's response:**

We thank the Referee for the careful revision and comments, which helped improving the overall quality of the manuscript.
A point-by-point answer to the referee's remarks is detailed in the following (in black the referee comments, in blue our answers, in green text modifications)

l.92: If you present molar yields, you must have information about the sensitivity of your mass spectrometer. In addition the y-axis in Figures 2 and 3 seem to be given in molecule concentration. (If not, that should be clarified in the captions.) At C1 other parts of the manuscript you mention that you cannot quantify dimers because the transmission of your TOF-MS is unknown. In this manuscript, any information about the sensitivity of your instrument is missing. How did you estimate the molar yields then? Please, state precisely in the experimental section, what you did to determine the sensitivity or what the basis of your assumptions is.

We answered this point in our reply to referee #1. The HOM molar yield was calculated from the ratio of HOM concentration to reacted ArHC (the yields were not corrected for HOMs losses in the flow tube). The amount of ArHC reacted was calculated based on the assumption that the OH production was similar in all experiments. The OH concentration was determined from the difference in precursor concentration between excimer lamp switched on or off from two D9-butanol, toluene, mesitylene, and biphenyl experiments. The HOMs quantification is based on the calibration factor for sulfuric acid and the assumption that HOMs have the same ionization efficiency as sulfuric acid (Ehn et al., 2014; Kirkby et al., 2016). It is known that the ionization efficiency depends on the structure of the HOMs. Since the ionization efficiency might be <1 for some HOM structures, the reported concentrations here represent lower limits. No further characterization of the system CIMS - ArHC HOMs was pursued. We added the following in the text:

HOMs yields are calculated as the ratio of HOMs measured to ArHC reacted. HOMs were quantified using the calibration factor for sulfuric acid and assuming the same charging efficiency for HOMs (Ehn et al., 2014; Kirkby et al., 2016). From the decrease of the precursor concentration (lights off versus lights on) we determined an average OH concentration. From the experiments with D9-butanol, toluene, mesitylene and biphenyl an average OH concentration of $2 \cdot 10^8$ OH cm$^{-3}$ was obtained. Assuming the same OH production of the lamp in all experiments ArHC reacted was calculated from the OH radical exposure using the reaction rate coefficients at 25°C.

Table 1 was complemented with HOMs concentration, reacted fraction of ArHCs and HOMs yield and looks now like this:

Table 1

Initial concentrations of precursors, reaction rate coefficients, ArHC reacted fraction (%), total HOMs concentration and HOMs yield (%) relative to the reacted ArHC. The mixing ratio of precursors was determined at the exit of the flow tube when the excimer lamp (OH generation) was switched off.

| Compound | Concentration (molecules cm$^{-3}$) | $k_{OH}$ ($10^{-12}$ cm$^3$ molecules$^{-1}$ s$^{-1}$) | Reacted fraction (%) | [HOM] (molecules cm$^{-3}$) | HOMs yield (%) |
|---|---|---|---|---|---|
| Benzene ($C_6H_6$) | $9.85 \cdot 10^{13}$ | 1.22 | 0.5 | $1.2 \cdot 10^9$ | 0.2 |
| Toluene ($C_7H_8$) | $1.97 \cdot 10^{13}$ | 5.63 | 2.3 | $4.4 \cdot 10^8$ | 0.1 |

| | | | | | |
|---|---|---|---|---|---|
| Ethylbenzene (C$_8$H$_{10}$) | 1.13 10$^{13}$ | 7.0 | 2.8 | 9.4 10$^8$ | 0.3 |
| (o/m/p)-xylene (C$_8$H$_{10}$) | 2.95 10$^{12}$ | 13.6/23.1/14.3 | // | 2.8 10$^9$ | // |
| Mesitylene (C$_9$H$_{12}$) | 2.46 10$^{12}$ | 56.7 | 22.7 | 3.1 10$^9$ | 0.6 |
| Naphthalene (C$_{10}$H$_8$) | 2.95 10$^{13}$ | 23.0 | 9.2 | 1.4 10$^{10}$ | 0.5 |
| Biphenyl (C$_{12}$H$_{10}$) | 4.43 10$^{13}$ | 7.1 | 2.8 | 1.8 10$^{10}$ | 1.4 |

Reference for the *k*-rates: (Atkinson and Arey, 2003)

p4, l.119: Does the split off of O2 explain the lower oxidation degrees of dimers assuming RO2+RO2 = ROOR +O2 or not?

We think this could be an explanation. We already give this explanation in line 122ff. To corroborate it we would need dedicated experiments which are beyond the scope of this paper.

p.4, l.119f and p.6, l.162: I would propose to give here Mentel et al. 2015 somewhat more recognition as they described, based on experimental observations in context of autoxidation, this type of dimer formation including mixed dimers a year before Kirkby et al. 2016. The same is true for the alkoxy path.

We added Mentel et al., (2015) instead of Kirkby et al. (2016) in the citations given in line 124 and line 162.

p.4, l.192ff: "Additionally, more oxygenated radicals have a higher probability to undergo an auto-termination radical reaction compared to a radical-radical recombination (RO2· + RO2· or RO2· + HO2·)." I don't exactly what you want to say with this statement in context of degree of methylation and dimer fraction. Less methylated aromatic compounds tend to more auto-termination? What means auto-termination - termination by internal reaction?

The referee seems to address our statement in line 124. Similar questions were also raised by referee #2. We report here our answer.

This statement can be derived from Rissanen 2014: We rewrote it:

Additionally, more oxygenated radicals have a higher probability to undergo a unimolecular termination compared to a radical-radical recombination (RO$_2$· + RO$_2$· or RO$_2$· + HO$_2$·. More oxygen atoms imply more peroxy functional groups and therefore a higher probability of a hydrogen abstraction in geminal position of a peroxide group which results in an OH radical loss and a carbonyl group formation. Therefore, the fraction of dimer formation should decrease with higher oxygen content.

p.8, l231ff: I think, that one should differentiate clearer between autoxidation by H-shift to peroxy radicals on one hand and by attack of the peroxy moiety to internal double bonds on the other hand. Although both reactions are internal rearrangements they are still of different character, as the first needs "mobile" H-atoms and the latter double bonds with potential to allyl radical formation. As a consequence the HOM formation in aromatic systems would be based - at least in parts- on a different mechanism?!

We agree with the referee. We rewrote this to read:

[revised manuscript text omitted]

---

## Referee Report (RR1)

Referee report on Molteni et al. " Formation of highly oxygenated organic molecules from aromatic compounds."

Major comments:

The language and the tone of voice have improved significantly from the previous version, which to my mind is in order considering the remaining open questions in the mechanistic details of the aromatic oxidation reactions. Now the results are presented more as an important new observation with less mechanistic details, which I think is adequate for the present. Thus, I can recommend this article for publication after few remaining issues, outlined below, have been duly addressed.

One particular, very recent publication would be good to inspect in this context: S. Wang et al. "Formation of Highly Oxidized Radicals and Multifunctional Products from the Atmospheric Oxidation of Alkylbenzenes" Environ Sci. Technol., 2017, 51, 8442. It is a joint theoretical-experimental description of "HOM" formation from alkylbenzenes. It seems there are some slight discrepancies between these studies, which could be worth mentioning in the discussion.

I am still having a bit hard time in understanding the used OH concentrations, and their influence on the oxidation system. The average OH concentration mentioned is $2 \times 10^8$ cm$^{-3}$, and this concentration should be enough for at least 3 OH attacks(!), which seems like a hard thing to accomplish in 20 second reaction time – to still yield a measurable product signal. Then in simulation of Appendix C a value approaching $10^{12}$ cm$^{-3}$ was used for initial OH concentration, although the main text gives the impression that all of the experiments were with the same photolysis source and power. In addition, it is said in the introduction that the OH reaction with aromatic system leads to "highly reactive products to more oxidants", and that the high reactivity is due to the non-aromatic double bonds. So it leads me wondering could the 140 ppb of ozone in the tube affect the secondary chemistry, if 3 OH attacks is possible too? Moreover, if 140 ppb $O_3$ was obtained at the setting with a concurrent $2 \times 10^8$ cm$^{-3}$ OH production, what was the corresponding value in the experiment with $8 \times 10^{11}$ cm$^{-3}$ initial OH? Could this discussion be made more consistent, and also, somehow clarified?

Minor comments:

1. Line 13: Wang et al. (ref given above) have published "HOMs" from alkylbenzenes.
2. Line 43: benzene is not the only aromatic ring, so the sentence needs some rewording.
3. Line 45. Same Wang et al. has shown experimentally and theoretically that HOM form from alkylbenzene oxidation reactions.
4. Line 46. Is Calvert 2002 the best reference for presenting carbon balance from oxidation of aromatic compounds? 50% seems like a very low number.
5. Line 58 onward: Could you state the purities of the compounds, and which were liquid and which solid.
6. Line 67: Could you shortly explain how a "HO2 generator" is suitable for OH generation. This does not come clear from the text. Also, as OH is much more reactive than HO2 I wonder if you needed to modify the generator?
7. Line 82: I do not understand why fluorinated compounds are mentioned if they do not play any role in the text.
8. Line 87: Multiplication sign missing.
9. Line 97: One dot too much.
10. Line 99: First n in m/z, or the biggest peaks summing up to 80%?
11. Line 220-222: I cannot follow how you get to 4 hydrogen more than the parent compound here.
12. Line 228: Usually aromatic rings are exceptionally stabile...

13. Line 229: I don't think "easier" is the right word here. I think "probable" would be closer to adequate wording.
14. Line 243-246: Why is "Type I" and "Type II" brought up in conclusions, but not in anywhere else?
15. Line 246: "Type II" autoxidation was already brought up (at least) in: Rissanen et al. J. Phys. Chem. A, 2015, 119, 4633, and Richters et al. Atmos. Chem. Phys., 16, 9831, 2016.
16. Line 250: Initial step of what?
17. Line 255 and 256: I think especially this finding could be discussed in light of results presented in Wang, S. et al 2017 (ref given above).
18. Line 270: Well, it is not a terribly big contribution. You suggested maximum HOM yield of 1.4%.
19. Table 1: Can you give a lumped value for the HOM yield from xylenes?
20. Table 2: It would be could to indicate which part of the products is assumed to be clusters, and which covalently bound molecules, as the naming "monomer, dimer, …" can be confusing here.
21. Appendix A: There are some HOMs that have even H but have a charge. How do you suppose these species were charged in the CIMS? Just curious here.
22. Appendix C: Perhaps add an opening paragraph explaining what is being presented?
23. Line 590: Would be nice to know more details of the mixing zone, and the time reaction mixture spends there in an ordinary experiment.
24. Line 590: Not "time in the lamp".
25. In Appendix C. The simulation uses a rather high value of OH. Why so?

---

## Author Response (AR2)

**Author's response:**

We thank the Referee for the careful revision and comments.

A point-by-point answer to the referees' remarks is given in the following (in black the referee comments, in blue our answers, in green text modifications)

The language and the tone of voice have improved significantly from the previous version, which to my mind is in order considering the remaining open questions in the mechanistic details of the aromatic oxidation reactions. Now the results are presented more as an important new observation with less mechanistic details, which I think is adequate for the present. Thus, I can recommend this article for publication after few remaining issues, outlined below, have been duly addressed.

One particular, very recent publication would be good to inspect in this context: S. Wang et al. "Formation of Highly Oxidized Radicals and Multifunctional Products from the Atmospheric Oxidation of Alkylbenzenes" Environ Sci. Technol., 2017, 51, 8442. It is a joint theoretical-experimental description of "HOM" formation from alkylbenzenes. It seems there are some slight discrepancies between these studies, which could be worth mentioning in the discussion.

We added the HOM formation pathway suggested by Wang et al., 2017 in our Figure 6 and compare our results with the theoretical expectations in the conclusions (see below).

I am still having a bit hard time in understanding the used OH concentrations, and their influence on the oxidation system. The average OH concentration mentioned is 2 x 108 cm-3, and this concentration should be enough for at least 3 OH attacks(!), which seems like a hard thing to accomplish in 20 second reaction time – to still yield a measurable product signal. Then in simulation of Appendix C a value approaching 1012cm-3 was used for initial OH concentration, although the main text gives the impression that all of the experiments were with the same photolysis source and power. In addition, it is said in the introduction that the OH reaction with aromatic system leads to "highly reactive products to more oxidants", and that the high reactivity is due to the non-aromatic double bonds. So it leads me wondering could the 140 ppb of ozone in the tube affect the secondary chemistry, if 3 OH attacks is possible too? Moreover, if 140 ppb O3 was obtained at the setting with a concurrent 2 x 108 cm-3 OH production, what was the corresponding value in the experiment with 8 x 1011 cm-3 initial OH? Could this discussion be made more consistent, and also, somehow clarified?

With the model we estimate the initial OH concentration to around  $8 \cdot 10^{11}$  cm-3. Since the OH radicals react rapidly the average OH concentration is much lower at around  $2 \cdot 10^8$  cm-3. This can be seen from Figure C1. As seen from Table 1 the fraction of precursor that reacted is a few percent. This results in roughly 10 ppb of oxidation products, which can potentially react again with an OH radical. This is not impossible. Some of these products will have a C=C double bond, however, a reaction with 140 ppb ozone is too slow to produce a measureable amount of products. We write now:

From the decrease of the precursor concentrations (lights off versus lights on) of D9-butanol, toluene, mesitylene and biphenyl we determined the fraction of reacted precursor. With a kinetic reaction model we then determined the initial OH concentration to be around  $8.5 \cdot 10^{11}$  cm-3. For the other precursors we assumed the same OH production of the lamp to calculate the fraction of reacted precursors (Table 1). Ozone, produced in the excimer irradiated region as a side product of the OH generation, was measured to be about 140 ppbv at the exit of the flow tube. It does not react with aromatic compounds.

Even though some OH-oxidation products will contain a C=C double bond, the slow reaction rate of these with ozone is not expected to form significant amounts of products via this route.

**Minor comments:**

1. Line 13: Wang et al. (ref given above) have published "HOMs" from alkylbenzenes. This was true at the time we submitted the discussion paper. Indeed, Wang et al have now also shown it.

We deleted the sentence stating that HOMs from alkylbenzenes were not observed yet.

2. Line 43: benzene is not the only aromatic ring, so the sentence needs some rewording.We do not claim that. We say "hydroxycyclohexadienyl-type" radical. This can also include substituents on the aromatic ring or polynuclear aromatics.We keep it as is.

3. Line 45. Same Wang et al. has shown experimentally and theoretically that HOM form from alkylbenzene oxidation reactions.

We added this reference in line 50.

Here we show the formation of HOMs from ArHC upon reaction with OH radicals as has also very recently been reported by Wang at al. (2017).

4. Line 46. Is Calvert 2002 the best reference for presenting carbon balance from oxidation of aromatic compounds? 50% seems like a very low number.

The numbers can vary significantly depending on the level of product identification one considers. We deleted this sentence as it is not relevant in the context.

5. Line 58 onward: Could you state the purities of the compounds, and which were liquid and which solid.

done

6. Line 67: Could you shortly explain how a "HO2 generator" is suitable for OH generation. This does not come clear from the text. Also, as OH is much more reactive than HO2 I wonder if you needed to modify the generator?

The lamp produces both OH and HO2 as one can see from the equations R-C1 to R-C11 in Appendix C. This reflects the ambient situation better than an OH-only source. We changed the text slightly to avoid confusion.

The Xe excimer lamp consists of a tubular quartz cell which surrounds a quartz flow tube (outer diameter 10 mm) (Bartels-Rausch et al., 2011). This light photolysises  $O_2$  and  $H_2O$  leading to the formation of OH and  $HO_2$  radicals (see Appendix C).

7. Line 82: I do not understand why fluorinated compounds are mentioned if they do not play any role in the text.

We think it is a useful information for experimentalists when using Nafion membranes for humidification. We would like to keep it.

8. Line 87: Multiplication sign missing. Text was changed to better explain OH concentration (see above)

9. Line 97: One dot too much. done

10. Line 99: First n in m/z, or the biggest peaks summing up to 80%? We summed up the largest n peaks. This is clarified now. ...for each compound we report the largest *n* peaks that sum up to 80% of the total detected signal of HOMs.

11. Line 220-222: I cannot follow how you get to 4 hydrogen more than the parent compound here. If we assume an OH radical that adds to an aromatic ring and one of the possible subsequent radical termination reactions (R3b, R4a) leading to a hydroperoxide (R3b) or an alcohol (R4a), we obtain in both cases a molecule with 2 hydrogen atoms more than the parent compound.

Assuming this molecule still contains a C=C double bond, a second OH radical can add to this double bond. Subsequent radical termination reactions as described above (R3b, R4a), can add two more hydrogen atoms to the molecular structure.

This leads to a compound that contains in total 4 hydrogen atoms more than the parent compound. We rewrote this section to better describe the formation of compounds with four hydrogen atoms more than the precursor. Furthermore, we included the mechanism of Wang et al. 2017 (comment 3) and addressed comment 14.

To this oxygen bridged bicyclic radical further oxygen additions seem to occur up to a peroxy radical with seven oxygen atoms (C9H13O7), which is the species detected at relatively high intensity (3.5%). Two potential routes are shown in Figure 6. One follows the traditional autoxidation mechanism with internal H-abstraction and oxygen addition (Type I autoxidation). The other route proposes another cyclization forming a second oxygen bridge. This mechanism also produces a carbon centered radical and promotes autoxidation (Type II) by the addition of another oxygen molecule. Wang et al. (2017) provide evidence from isotope labelling experiments for the occurrence of Type I autoxidation in isopropylbenzene. Some of the ArHC tested also form radicals with a higher number of odd oxygens (i.e., up to 9-11 O atoms, Appendix B) indicating that autoxidation may even proceed further. Compounds with an even number of oxygen are formed via the alkoxy pathway and may also include a ring opening step. Possible branching channels where this may happen are indicated in Figure 6.

Termination reactions to alcohols (R3b) or hydroperoxides (R4a) can form molecules still containing double bonds which can further add an OH radical leading to compounds with four hydrogen atoms more than the precursor.

12. Line 228: Usually aromatic rings are exceptionally stabile...

We say that in second generation the other aromatic ring can also be attacked by an OH radical. The OH concentration in our system is high and this is kinetically possible. We do not see another explanation

for the compounds with a high number of H-atoms. We rephrase the sentence on line 229 (see the following referee question 13)

13. Line 229: I don't think "easier" is the right word here. I think "probable" would be closer to adequate wording.

Due to the high OH concentration a second OH attack is probable.

14. Line 243-246: Why is "Type I" and "Type II" brought up in conclusions, but not in anywhere else? We introduce this now already in the discussion of the mechanism (Figure 6) together with the pathway as suggested by Wang et al. 2017 (see answer to comment 11).

15. Line 246: "Type II" autoxidation was already brought up (at least) in: Rissanen et al. J. Phys. Chem. A, 2015, 119, 4633, and Richters et al. Atmos. Chem. Phys., 16, 9831, 2016.

We changed the sentence to:

Type II autoxidation has also been proposed to occur with sesquiterpenes with two double bonds (Richter et al. 2016) and also with  $\alpha$ -pinene for certain HOMs (Rissanen et al., 2015).

16. Line 250: Initial step of what?

We changed the sentence.

Recent studies (Nakao et al., 2011; Schwantes et al., 2016) suggest that from hydroxy ArHC equivalents, which are formed in the first-generation OH oxidation of aromatics, additional OH oxidation steps can produce substantial amounts of polyhydroxy aromatics with high O:C ratio.

17. Line 255 and 256: I think especially this finding could be discussed in light of results presented in Wang, S. et al 2017 (ref given above).

This is the wrong place to discuss Wang et al. Here, we want to show that the formation of polyhydroxy aromatics does not fit our data.

We discuss now Wang et al. in a subsequent section:

Quantum chemical calculations have revealed that intramolecular H-migrations in bicyclic peroxy radicals may be a feasible route to HOMs (Wang et al., 2017). Especially aromatics with a longer chain substituent (ethyl-, isopropylbenzene) or multiple substituents may have a fast HOM formation pathway. Indeed, our measurements show somewhat higher HOM yields for xylene and mesitylene compared to benzene and toluene. However, more kinetic and mechanistic studies are needed to better understand HOM formation from the various aromatics.

18. Line 270: Well, it is not a terribly big contribution. You suggested maximum HOM yield of 1.4%. We deleted this sentence.

19. Table 1: Can you give a lumped value for the HOM yield from xylenes?

We give a range of HOM yields assuming that the xylene mixture is composed only of one xylene isomer with either the slowest or fasted OH reaction rate constant. We updated the yields based on the model calculation which was not done in the last version.

20. Table 2: It would be could to indicate which part of the products is assumed to be clusters, and which covalently bound molecules, as the naming "monomer, dimer, …" can be confusing here. We cannot distinguish between these two types. It would require an instrument that is able to separate these, as e.g an IM-MS.

21. Appendix A: There are some HOMs that have even H but have a charge. How do you suppose these species were charged in the CIMS? Just curious here. We do not know and can only speculate.

22. Appendix C: Perhaps add an opening paragraph explaining what is being presented? We added:

Here a description of the OH generator and the involved reactions is given. A kinetic reaction model was developed to investigate the effect of uncertainties in the initial OH concentration on the oxidation product distribution.

23. Line 590: Would be nice to know more details of the mixing zone, and the time reaction mixture spends there in an ordinary experiment.

We have not evaluated this because we did not do kinetic experiments. The goal of the experiments was to investigate the potential formation of HOMs from aromatics and differences in mechanisms. The residence time in the flow tube is 20 s. We estimate that the mixing in the beginning of the flow tube is on the order of 1-2 seconds. We do not think it is useful to add here some speculation which is not relevant for the results of the paper.

24. Line 590: Not "time in the lamp".

We modified:

The final OH concentration entering the mixing zone depends on the residence time of the air in the lamp and in the transfer region.

25. In Appendix C. The simulation uses a rather high value of OH. Why so?

This is the value we derived from the experiment. It is stated in the text: The initial OH radical concentration  $(8.5 \cdot 10^{11} \text{ molecules cm}^{-3})$  is tuned in order to match the OH exposure, which was determined from the amount of reacted mesitylene.

**Formation of highly oxygenated organic molecules from aromatic compounds.**

Ugo Molteni1, Federico Bianchi2, Felix Klein1, Imad El Haddad1, Carla Frege1, Michel J. Rossi1, Josef Dommen1, Urs Baltensperger1,\*

[revised manuscript text omitted]
                 | k OH                                                 | Reacted fraction         | [HOM]                         | HOMs yield               |
|--------------------------------------------------|-------------------------------|-----------------------------------------------------------------|--------------------------|-------------------------------|--------------------------|
|                                                  | (molecules cm -3 ) | $(10^{-12} \text{ cm}^3 \text{ molecules}^{-1} \text{ s}^{-1})$ | (%)                      | (molecules cm -3 ) | (%)                      |
| Benzene (C 6 H 6 )         | 9.85 10 13         | 1.22                                                            | 0.5                      | 1.2 10 9           | 0.2                      |
| Toluene (C 7 H 8 )         | 1.97 10 13         | 5.63                                                            | 2.3                      | 4.4 10 8           | 0.1                      |
| Ethylbenzene (C 8 H 10 )   | 1.13 10 13         | 7.0                                                             | 4.4 2.8           | 9.4 10 8           | 0. 2 3            |
| (o/m/p)-xylene (C 8 H 10 ) | 2.95 10 12         | 13.6/23.1/14.3                                                  | # 5.4 - 9.2       | 2.8 10 9           | 1.0 - 1.7 #       |
| Mesitylene (C 9 H 12 )     | 2.46 10 12         | 56.7                                                            | 22.7                     | 3.1 10 9           | 0.6                      |
| Naphthalene (C 10 H 8 )    | 2.95 10 13         | 23.0                                                            | 2.6 9.2           | 1.4 10 10          | 1.8 0.5           |
| Biphenyl (C 12 H 10 )      | 4.43 10 13         | 7.1                                                             | <del>2.81.6</del> | 1.8 10 10          | <del>1.42.5</del> |

495

Reference for the k-rates: Atkinson and Arey, 2003.

**Table 2**

500

Summary of HOM characteristics. For each of the 7 compounds the percentage fractional distribution of the signal is presented. For monocyclic compounds the distribution comprises monomers and dimers, for naphthalene and biphenyl monomers, dimers, trimers and tetramers are reported. These values are not quantitative as the instrument cannot be calibrated for such compounds. For each band the weighted arithmetic means of the O:C ratio are reported in parentheses. The fraction of the identified peaks as adduct with  $NO_3^-$  is given in the last column.

| Common 1                                       | Bands distribution |             |              |                |                                    |  |
|------------------------------------------------|--------------------|-------------|--------------|----------------|------------------------------------|--|
| Compound                                       | Monomer (O:C)      |             | Dimer (O:C)  |                | Adduct ( $\Pi O M \bullet N O_3$ ) |  |
| Benzene (C 6 H 6 )       | 80 (1.08)          |             | 20 (0.91)    |                | 0.91                               |  |
| Toluene (C 7 H 8 )       | 71 (1.09)          |             | 29 (0.75)    |                | 0.94                               |  |
| Ethylbenzene (C 8 H 10 ) | 69 (0.86)          |             | 31 (0.62)    |                | 0.83                               |  |
| $(o/m/p)$ -xylene $(C_8H_{10})$                | 65 (0.78)          |             | 35 (         | 0.57)          | 0.92                               |  |
| Mesitylene (C 9 H 12 )   | 61 (0.81)          |             | 39 (0.49)    |                | 0.92                               |  |
|                                                | Monomer (O:C)      | Dimer (O:C) | Trimer (O:C) | Tetramer (O:C) |                                    |  |
| Naphthalene (C 10 H 8 )  | 34 (0.55)          | 64 (0.29)   | 2 (0.34)     | 1 (0.28)       | 0.84                               |  |
| Biphenyl (C 12 H 10 )    | 52 (0.44)          | 43 (0.35)   | 4 (0.29)     | 1 (0.32)       | 0.77                               |  |

505

**Figures**

510

Figure 1. Experimental set-up. Zero air from a pure air generator is split into 3 flows. A sheath flow of 6.7 L min-1. An air stream of 1.1 L min-1 collects vapors from a reagent compound vial and is then mixed with a humidified air stream of 7 L min-1 (RH 75%) which carries OH free radicals generated through irradiation at 172 nm.